# Selective sorting of ancestral introgression in maize and teosinte along an elevational cline

**Erin Calfee**[1,2]*, **Daniel Gates**[2¤c], **Anne Lorant**[3¤a], **M. Taylor Perkins**[2¤b], **Graham Coop**[1,2‡], **Jeffrey Ross-Ibarra**[1,2,4‡]

**1** Center for Population Biology, University of California, Davis, California, United States of America,
**2** Department of Evolution and Ecology, University of California, Davis, California, United States of America,
**3** Department of Plant Sciences, University of California, Davis, California, United States of America,
**4** Genome Center, University of California, Davis, California, United States of America

¤a Current address: Laboratoire de Biologie Moléculaire et Cellulaire du Cancer, Hôpital Kirchberg, Luxembourg
¤b Current address: Department of Biology, Geology, and Environmental Science, The University of Tennessee at Chattanooga, Tennessee, United States of America
¤c Current address: Checkerspot, Inc., Berkeley, California, United States of America
‡ These authors are co-mentor authors on this work.
* erincalfee@gmail.com

**Data Availability Statement:** Raw Illumina sequence data generated by this study is available through the NCBI Short Read Archive, PRJNA657016 (maize/mexicana) and SRR7758238 (*Tripsacum*). All relevant metadata is

## Abstract

While often deleterious, hybridization can also be a key source of genetic variation and pre-adapted haplotypes, enabling rapid evolution and niche expansion. Here we evaluate these opposing selection forces on introgressed ancestry between maize (*Zea mays* ssp. *mays*) and its wild teosinte relative, *mexicana* (*Zea mays* ssp. *mexicana*). Introgression from ecologically diverse teosinte may have facilitated maize's global range expansion, in particular to challenging high elevation regions (> 1500 m). We generated low-coverage genome sequencing data for 348 maize and *mexicana* individuals to evaluate patterns of introgression in 14 sympatric population pairs, spanning the elevational range of *mexicana*, a teosinte endemic to the mountains of Mexico. While recent hybrids are commonly observed in sympatric populations and *mexicana* demonstrates fine-scale local adaptation, we find that the majority of *mexicana* ancestry tracts introgressed into maize over 1000 generations ago. This *mexicana* ancestry seems to have maintained much of its diversity and likely came from a common ancestral source, rather than contemporary sympatric populations, resulting in relatively low $F_{ST}$ between *mexicana* ancestry tracts sampled from geographically distant maize populations.

Introgressed *mexicana* ancestry in maize is reduced in lower-recombination rate quintiles of the genome and around domestication genes, consistent with pervasive selection against introgression. However, we also find *mexicana* ancestry increases across the sampled elevational gradient and that high introgression peaks are most commonly shared among high-elevation maize populations, consistent with introgression from *mexicana* facilitating adaptation to the highland environment. In the other direction, we find patterns consistent with adaptive and clinal introgression of maize ancestry into sympatric *mexicana* at many loci across the genome, suggesting that maize also contributes to adaptation in *mexicana*, especially at the lower end of its elevational range. In sympatric maize, in addition to high introgression regions we find many genomic regions where selection for local adaptation

found within the Supporting information files. Access to previously published genomic resources used in this study: B73 maize reference genome v4 (Gramene), recombination map from Ogut et al. 2015 (https://www.panzea.org/publications), maize reference panel sequences (NCBI SRA PRJNA616247), and parviglumis reference panel sequences (NCBI SRA PRJNA616247). Local ancestry posterior probability files, ancestry_hmm input files, genomewide ancestry estimates, and summarised population allele frequency data are accessible via Figshare (https://doi.org/10.6084/m9.figshare.16641799). Scripts are available at https://github.com/ecalfee/hilo.

**Funding:** This work was funded by the the Division of Integrative Organismal Systems from the National Science Foundation, www.nsf.gov (NSF No. 1546719, awarded to JRI and GC), the National Institute of General Medical Sciences of the National Institutes of Health, www.nigms.nih.gov (NIH R01 GM108779 and R35 GM136290, awarded to GC), and the University of California, Davis, https://ucdavis.edu (Loomis Graduate Fellowship in Agronomy, awarded to EC). The funders had no role in the study design, data collection and analysis, decision to publish, or preparation of the manuscript.

**Competing interests:** The authors have declared that no competing interests exist.

maintains steep gradients in introgressed *mexicana* ancestry across elevation, including at least two inversions: the well-characterized 14 Mb *Inv4m* on chromosome 4 and a novel 3 Mb inversion *Inv9f* surrounding the *macrohairless1* locus on chromosome 9. Most outlier loci with high *mexicana* introgression show no signals of sweeps or local sourcing from sympatric populations and so likely represent ancestral introgression sorted by selection, resulting in correlated but distinct outcomes of introgression in different contemporary maize populations.

## Author summary

When species expand their ranges, new encounters with diverse wild relatives can introduce deleterious genetic variation, but may also accelerate the colonization of novel environments by providing 'ready-made' genetic adaptations. Maize today is a global staple, far exceeding the original ecological niche of its wild progenitor. We show that gene flow from highland-endemic wild *mexicana* facilitated maize's range expansion from the valleys where it was domesticated to sites over 1500m in the mountains of Mexico. We find loci where *mexicana* ancestry has been repeatedly favored in highland maize populations. We also find loci (including a newly identified inversion) where *mexicana* ancestry increases steeply with elevation, providing evidence for adaptive trade-offs.

We additionally demonstrate selection against *mexicana* ancestry, especially near domestication genes. We sampled *mexicana* growing alongside maize fields, yet find little evidence that introgression is recent or locally-sourced genomewide or at adaptive loci. Rather, the majority of *mexicana* ancestry was introduced into maize over 1000 generations ago, and subsequently diverged and was sorted by selection in individual populations. These results add to our understanding of the effects of introgression on range expansions and adaptation.

## Introduction

Interbreeding between partially diverged species or subspecies can result in admixed individuals with low fitness, e.g. due to hybrid incompatibilities [1–3]. Consistent with the view that hybridization is often deleterious, a growing number of species show evidence of pervasive selection against introgressed ancestry [4–13]. At the same time, introgression can be a source of novel genetic variation and efficiently introduce haplotypes carrying sets of locally adapted alleles, with the potential for rapid adaptation to new ecological challenges [14]. Indeed, admixture has been linked to adaptive species radiations and/or rapid niche expansions in a number of natural systems, including mosquitoes [15], *Drosophila* [16], butterflies [9], cichlids [17], sunflowers [18], wild tomatoes [19] and yeast [20, 21]. In addition, introgression from wild relatives has facilitated the broad range expansions of multiple domesticated crops (reviewed in [22] and [23]), and gene flow from crops back into their wild relatives has in some cases opened up novel 'weedy' niches [24].

Maize (*Zea mays* ssp. *mays*) is an ideal system to study selection on admixed ancestry and the effects on range expansion, as it has colonized nearly every human-inhabited ecosystem around the world [25] and interbreeds with a number of wild relatives genetically adapted to distinct ecologies [26, 27]. In Mexico, highland maize represents an early major niche expansion that may have been facilitated by introgression. Approximately 9 thousand years ago,

maize (*Zea mays* ssp. *mays*) was domesticated in the Balsas River Valley in Mexico from a lowland-adapted sub-species of teosinte (*Zea mays* ssp. *parviglumis* [28]), which grows readily at sea level and lower elevations of the Sierra Madre del Sur [29]. In contrast, *Zea mays* ssp. *mexicana*, which diverged from *parviglumis* about 60 thousand years ago [30], is endemic to highland regions in Mexico (∼1500–3000 meters in elevation) where it has adapted to a number of ecological challenges: a cooler, drier climate with higher UV intensity, different soil nutrient composition, and a shorter growing season necessitating earlier flowering times [31–35].

Maize was introduced as a crop to the mountains of Mexico around 6.2 thousand years ago [36], and it is thought that gene flow from *mexicana* assisted in adaptation to high elevation selection pressures. Highland maize and *mexicana* share a number of putatively adaptive phenotypes [37, 38], including earlier flowering times for the shorter growing season [34], purple anthocyanin-based pigmentation which shields DNA from UV damage [39] and increases solar heat absorption [40], and macrohairs on the leaf and stem sheath, which are thought to increase herbivore defense [41] and/or heat maintenance in colder environments [42]. Earlier studies using 50K SNP-chip data for highland populations [43] or genomewide data for a small number of individuals [44, 45], have shown that highland maize populations have experienced significant admixture from *mexicana*, reaching high frequency at some loci, consistent with adaptive introgression.

While some highland and locally-adapted alleles may be beneficial to maize, many introgressed *mexicana* alleles, especially those affecting domestication traits, should be selected against by farmers growing maize. In addition, maize alleles introgressed into *mexicana* should be selected against because maize has accumulated genetic load from reduced population sizes during domestication [44] and because domestication traits generally reduce fitness in the wild [46–48], e.g. loss of disarticulation and effective seed dispersal [37].

In this study, we generate whole genome sequencing data to investigate genomic signatures of admixture and selection in paired sympatric maize and *mexicana* populations, sampled from 14 locations across an elevational gradient in Mexico. *Mexicana* was sampled from wild populations and maize was sampled from nearby fields where traditional cultivation methods and open pollination have resulted in populations with distinct local characteristics and high phenotypic and genetic diversity (often called maize 'landraces'). This expanded sampling of sympatric maize and *mexicana* populations across Mexico, combined with genomewide data and a well-parameterized null model, improves our ability to more formally test for adaptive introgression and identify likely source populations. The source of introgression is of interest, as teosinte demonstrates local adaptation to different niches within the highlands and there is significant genetic structure between *mexicana* ecotypes [32, 37, 49–51]. Thus we can test whether local *mexicana* populations are the ongoing source for geographically-restricted locally adaptive haplotypes. We use this comprehensive genomic dataset to characterize the bi-directional timing and origin of introgression and evaluate the patterns and scale of natural selection for and against admixture between these taxa.

## Results/discussion

### Genomewide *mexicana* ancestry is structured by elevation

We sampled paired sympatric populations from 14 geographically dispersed locations to assess the extent of gene flow between maize and *mexicana* in Mexico. Maize today is grown across the entire elevational range of its wild teosinte relatives, from sea-level up to 4000 meters [52]. Our sampled sites range from 1547–2600 meters in elevation, which spans a large portion of *mexicana*'s range and exceeds the upper elevational range for maize's wild ancestor,

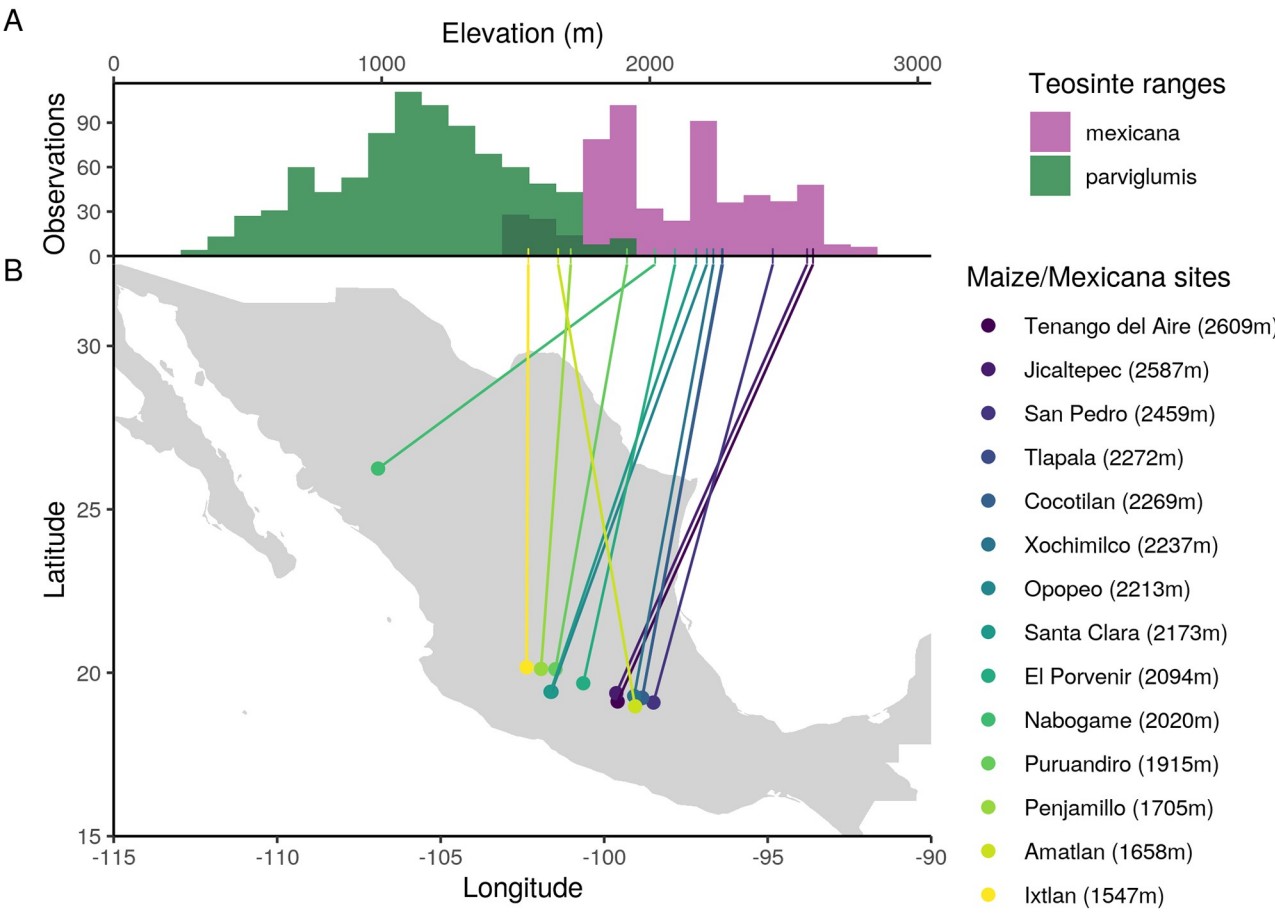

**Fig 1. Sampled sympatric maize/*mexicana* populations compared to the distribution of teosintes.** (A) Elevational range of teosintes based on historical occurrence data (1842–2016) from [29]. *Parviglumis* and *mexicana* overlap at middle elevations (dark green) and maize today is grown across this entire elevational range. (B) Geographic location and elevation of contemporary sympatric maize and *mexicana* population pairs sampled across 14 sites. Map of Mexico created with Natural Earth data (https://www.naturalearthdata.com).

*parviglumis* (Fig 1). For each of 14 maize/*mexicana* sympatric sample locations, we resequenced 7–15 individuals per subspecies.

We additionally sequenced 43 individuals from 3 *mexicana* reference populations, totalling 348 low-coverage genomes (mean ∼1x). Two of these *mexicana* reference populations are documented to have no adjacent maize agriculture within the past 50 years, while a third higher elevation population (Amecameca) was chosen because it grows above the elevational range of *parviglumis*, and thus outside of the historical range of maize. For a maize reference population, we added 55 previously published high-coverage genomes from a population grown near Palmar Chico at 983 m [53, 54], well below the elevational range of *mexicana*. Because *parviglumis* is known to admix historically with both of our focal subspecies in Mexico, we also included 50 previously published high-coverage *parviglumis* genomes from the 'Mound' population at 1,008 m, also near Palmar Chico [53–55]. Completely allopatric reference populations are not available because maize has been grown at high density throughout Mexico across the full elevational range of both teosintes. *A priori*, gene flow from maize into *mexicana* is possible at Amecameca, and historically between maize and teosinte at all locations. We therefore assess for possible gene flow into each reference population below.

Principal components analysis of genetic diversity clearly separates maize and *mexicana*, with putative admixed individuals from sympatric populations having intermediate values along PC1. Additionally, PC2 provides evidence of gene flow from *parviglumis*, particularly into lower elevation *mexicana* populations (S1 Fig), which motivated us to analyse a 3-way admixture scenario in all subsequent analyses.

To estimate genomewide ancestry proportions for each individual, we ran NGSAdmix [56] with K = 3 genetic clusters and genotype likelihoods for all *mexicana*, maize, and *parviglumis* individuals. The three genetic clusters clearly map onto *mexicana*, maize and *parviglumis* ancestry, with no significant admixture in the lowland maize or *parviglumis* reference populations. We find minority *parviglumis* ancestry in the two lower-elevation reference *mexicana* populations, but no evidence of introgression from *parviglumis* or maize into the highest elevation population at Amecameca (Fig 2A).

Looking across samples from the 14 sympatric sites, we find a positive association between ancestry proportion and elevation (km), with higher *mexicana* ancestry at higher elevations in both sympatric maize ($\beta = 0.22$, $P = 1.01 \times 10^{-31}$) and sympatric *mexicana* ($\beta = 0.32$, $P = 2.02 \times 10^{-38}$) individuals (Fig 2B).

Increasing *mexicana* ancestry at higher elevations is consistent with selection favoring *mexicana* ancestry at higher elevations, but could also be due to purely demographic processes, e.g. a higher density of (wind-dispersed) *mexicana* pollen at higher elevations, or increased gene flow from non-admixed maize populations at lower elevations. While most populations have admixture proportions well-predicted by their elevation, outlier populations may be the result of recent colonization histories for some locations or adaptation to other environmental niches. Within teosintes, elevation is a major axis of niche separation between *parviglumis* (the ancestor of maize) and *mexicana* [50, 57], but genetic differentiation also correlates with soil nutrient content and at least four principal components constructed from climatic variables [33].

## Origin and timing of introgression

If *mexicana* ancestry found in contemporary maize genomes facilitated maize's colonization of the highlands approximately 6.2 thousands years ago [36], we would expect introgressed ancestry tracts to be short, due to many generations of recombination, and possibly to be derived from an ancient source population common to many present-day maize populations. To test these predictions, we estimated local ancestry across the genome for individuals from each sympatric maize and *mexicana* population using a hidden Markov model (HMM) based on read counts ([58]; see Materials and methods). For each admixed population, this HMM simultaneously estimates local ancestry and, by optimizing the transition rate between different (hidden) ancestry states, the generations since admixture. We assumed a 3-way admixture scenario in which a founding *mexicana* population receives a pulse of *parviglumis* ancestry, then a pulse of maize ancestry.

Admixture between maize and *mexicana* is generally old, with median estimates of 1014 generations for sympatric maize populations and 509 generations for sympatric *mexicana* populations (S3 Fig). Parviglumis admixture timing estimates vary substantially across populations (median: 173, range: 29–1006). Because the HMM fits a single-pulse per ancestry to what was almost certainly multiple admixture events over time, we caution against over-interpretation of exact dates. Multiple pulses or ongoing gene flow biases estimates towards the more recent pulse(s) [59, 60] and even old estimates do not exclude the possibility of limited more recent admixture. These single-pulse approximations do, however, provide evidence that a large proportion of the introgression, especially *mexicana* ancestry into maize, is found on short ancestry tracts and therefore relatively old.

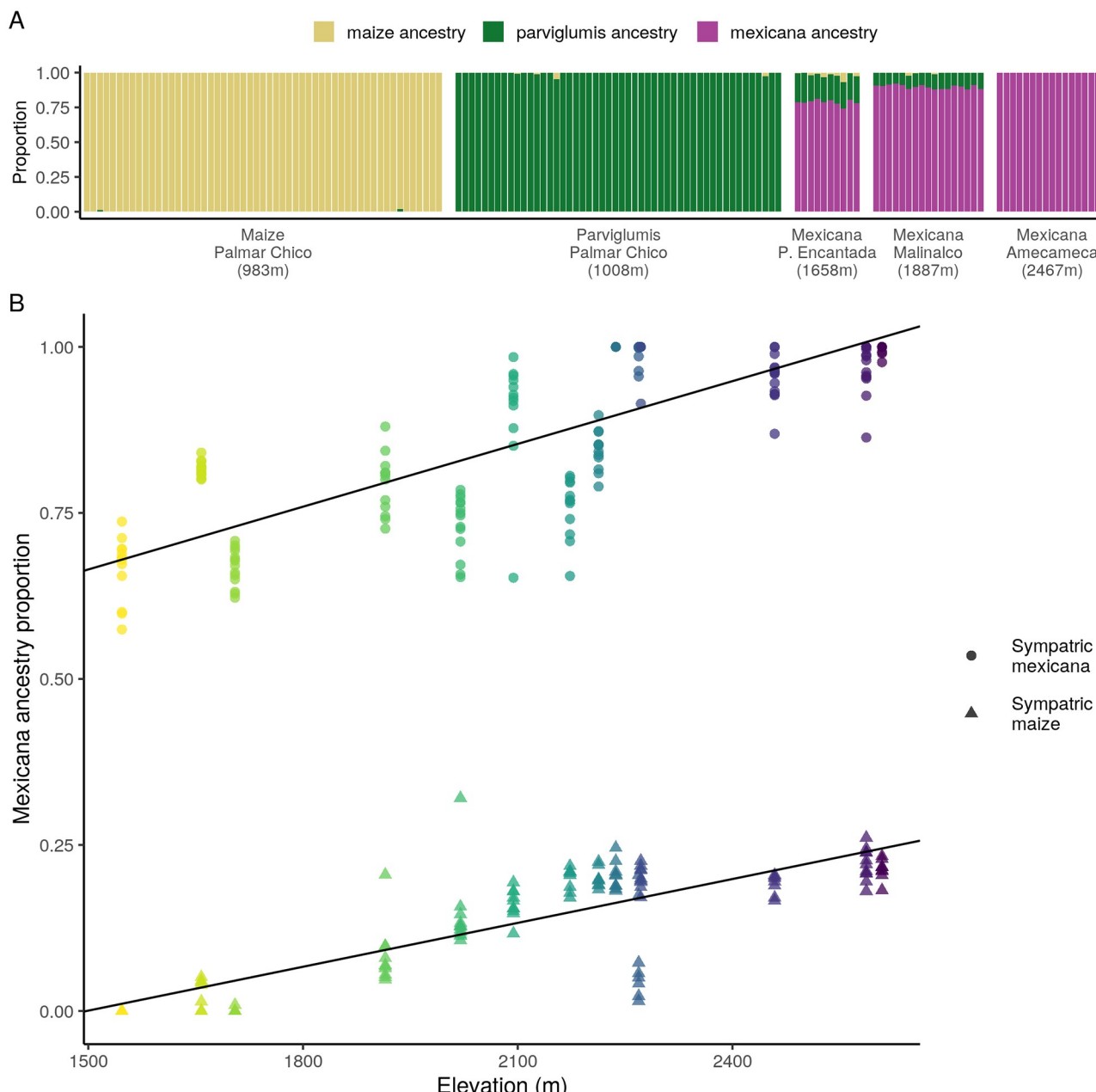

**Fig 2. Distribution of *mexicana* ancestry by elevation.** (A) Genomewide ancestry estimates (NGSAdmix) for reference maize, *mexicana* and *parviglumis* individuals, grouped by sampling location. (B) Genomewide *mexicana* ancestry estimates (NGSAdmix) for sympatric maize and *mexicana* individuals (n = 305) along an elevational gradient, colored by sampling location. Lines show best linear model fit for *mexicana* ancestry by elevation for each subspecies separately.

To identify likely source population(s) for introgressed ancestry, we compared $F_{ST}$ between all sympatric populations using only reads from high-confidence homozygous ancestry tracts (posterior > 0.8) for maize and *mexicana* ancestry separately. We find that most *mexicana* ancestry in maize resembles other *mexicana* ancestry introgressed into other maize populations, rather that *mexicana* ancestry from the local sympatric *mexicana* population (Fig 3).

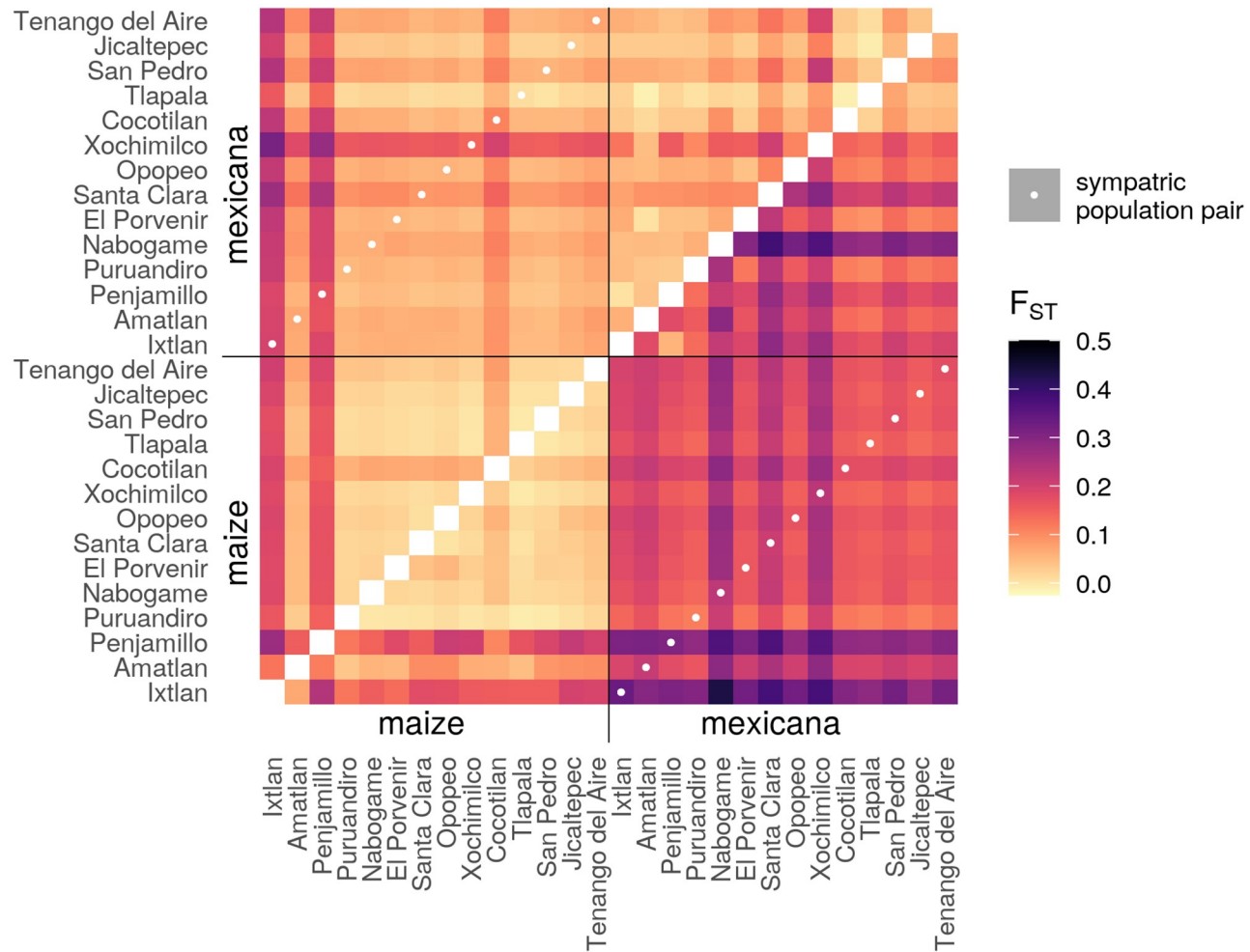

**Fig 3. $F_{ST}$ between ancestry tracts from different populations.** $F_{ST}$ between each pair of populations for maize ancestry tracts are shown in the upper left triangle, while $F_{ST}$ estimates for *mexicana* ancestry tracts are shown in the lower right triangle. Populations are sorted by subspecies, then elevation. Local sympatric maize-*mexicana* population pairs are highlighted with a white dot and do not show reduced $F_{ST}$ relative to other (non-local) maize-*mexicana* comparisons. Additionally, introgressed *mexicana* ancestry shows low differentiation between maize populations (creating a light-colored maize block in the left corner of the lower right triangle) and no potential *mexicana* source populations show especially low $F_{ST}$ with this block. Light coloring generally across the upper left triangle reflects the low differentiation within maize ancestry, providing little information to distinguish between potential maize ancestry sources.

This finding is consistent with most introgressed ancestry being drawn from a communal source population, but none of the sympatric *mexicana* populations have low enough $F_{ST}$ to tracts introgressed into maize to be a recent source. While we cannot rule out recent introgression from an unsampled source population, the timing of our admixture estimates is more consistent with divergence of *mexicana* ancestry, once introgressed into a maize background, from its original source population(s) (S3 Fig). Additionally, *mexicana* ancestry tracts in maize have only slightly reduced genetic diversity ($\pi$, S4 Fig), meaning many *mexicana* haplotypes have introgressed into maize at any given locus, with no evidence of a strong historical bottleneck.

Two lower elevation maize populations are an exception to this general pattern: Ixtlan and Penjamillo. These populations have higher $F_{ST}$ between their introgressed ancestry tracts and other *mexicana* tracts in maize (Fig 3), more recent timing of admixture estimates (S3 Fig),

and reduced genetic diversity (S4 Fig). These patterns could be caused by small population sizes and more recent independent admixture, although $F_{ST}$ does not identify a likely *mexicana* source population. Consistent with this interpretation, we have evidence that local maize at Ixtlan is at least partially descended from recently introduced commercial seed (relayed by local farmers [43]).

The lack of a clear reduction in $F_{ST}$ for *mexicana* ancestry tracts between sympatric population pairs, combined with older timing of admixture estimates, indicates that while contemporary hybridization may occur in the field between maize crops and adjacent *mexicana* populations, this is not the source for the bulk of the introgressed *mexicana* ancestry segregating in highland maize.

Instead, we propose that the majority of *mexicana* ancestry in maize derives from admixture over 1000 years ago, possibly from a diverse set of *mexicana* source populations over a large geographic and temporal span, and the resulting ancestry tracts are now distributed across different contemporary maize populations. These genomewide average $F_{ST}$ results, however, do not exclude the possibility that adaptively introgressed haplotypes at a particular locus came from one or more distinct, possibly local, source populations. While we also analyzed $F_{ST}$ within high-confidence maize ancestry tracts, we found that maize ancestry is too homogeneous to make inferences about potential admixture source populations of maize into *mexicana* (Fig 3 and S4 Fig).

## Selection against introgression genomewide

When there is widespread selection against introgressing variants at many loci across the genome, selection will more efficiently remove linked ancestry in regions of the genome with lower recombination rates, which creates a positive relationship between local recombination rate and the proportion of introgressed ancestry [4–13, 61]. To test whether such negative selection is shaping patterns of introgression genomewide in sympatric maize and *mexicana*, we first divided the genome into quintiles based on the local recombination rates for 1 cM windows. We then ran NGSAdmix on the SNPs within each quintile separately, using K = 3 clusters, to estimate ancestry proportions for each quintile. We used a recombination map from maize [62], which is likely to be correlated with other *Zea* subspecies at least at the level of genomic quintiles. A limitation of this analysis, however, is that we do not have a recombination map for hybrid populations, which means that e.g. segregating structural inversions will not necessarily show low recombination rates.

Our results from sympatric maize are consistent with selection against *mexicana* introgression at many loci genomewide, resulting in lower introgressed ancestry in regions of the genome with lower recombination rates (Fig 4A). We find a positive Spearman's rank correlation between recombination rate quintile and mean introgressed *mexicana* ancestry proportion ($\rho$ = 1.00, CI$_{95}$[0.80, 1.00]), reflecting the fact that introgression increases monotonically across quintiles. A similar analysis using $f_4$ statistics replicates this result (see Materials and methods, S5 and S6 Figs). The higher elevation maize populations show this pattern most starkly; while all individuals have low *mexicana* ancestry for the lowest recombination rate quintile, some high elevation populations have individuals with over 40% introgressed ancestry for the highest recombination rate quintile (Fig 4B). Using a linear-model fit, we found a significant positive interaction between recombination rate quintile and the slope of ancestry across elevation in sympatric maize (S4 Table). This is again consistent with low-recombination rate regions having a stronger effect of linked selection reducing *mexicana* ancestry, with higher elevation maize populations either experiencing larger amounts of gene flow or retaining more ancestry due to adaptive processes in high recombination regions.

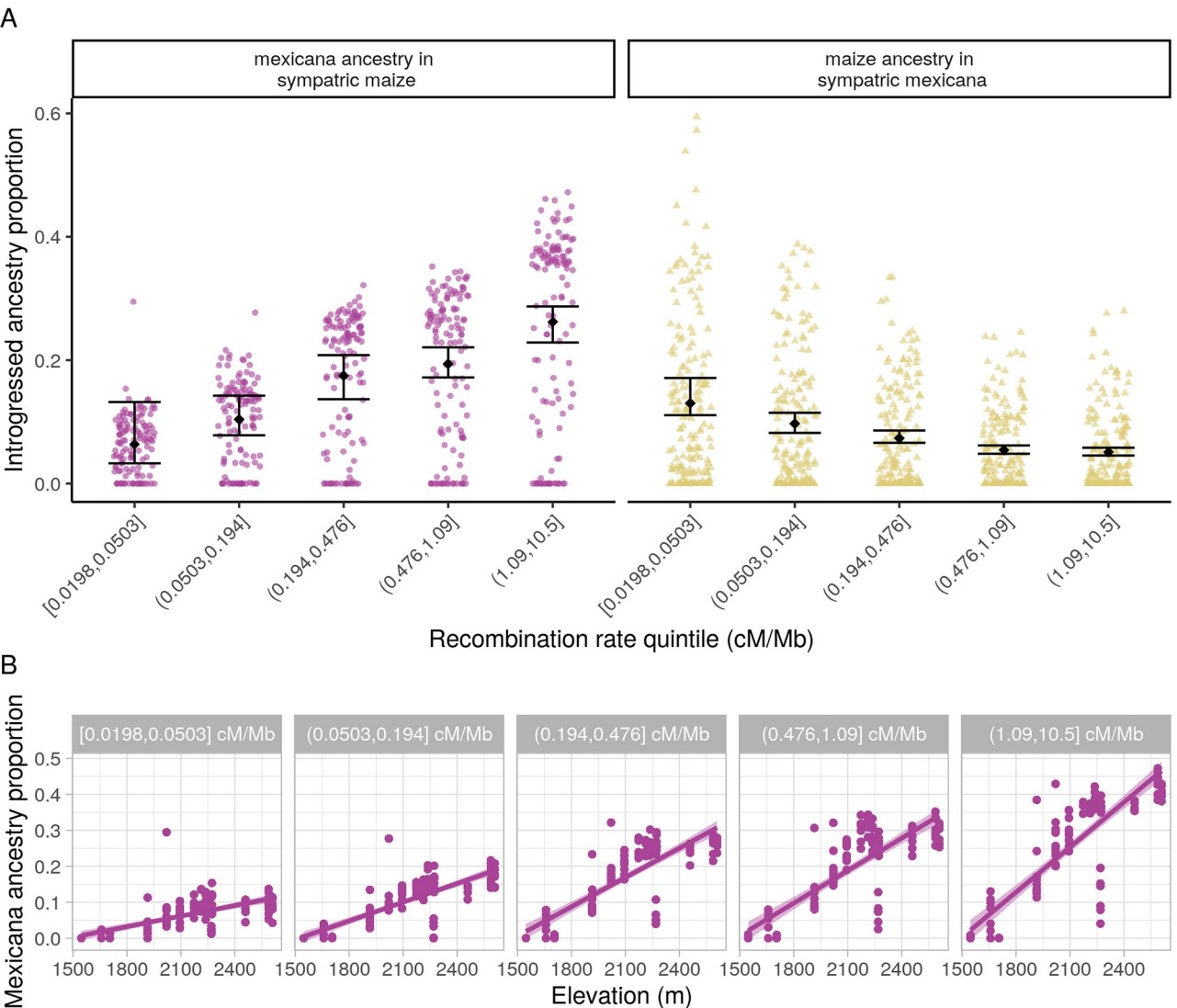

**Fig 4. . (A) Introgressed ancestry by recombination rate**. Inferred average genomewide introgressed ancestry in sympatric maize and *mexicana* individuals (NGSAdmix K = 3), by recombination rate quintiles. Group mean and 95% confidence interval based on bootstrap percentiles (n = 100) are depicted in black. Introgressed ancestry estimates for each individual are shown as points and points are jittered for better visualization. (B) Slope of *mexicana* ancestry introgressed into maize populations across elevation for each recombination rate quintile, based on NGSAdmix estimates. Each point is a sympatric maize individual and lines show the best-fit linear model for ancestry by elevation (with shaded 95% confidence interval), estimated separately for each quintile.

Because recombination rate is positively correlated with gene density in *Zea* [63], we also tested the Spearman's rank correlation between quintiles defined by coding base pairs per cM and their proportion introgressed *mexicana* ancestry. Again we found evidence supporting pervasive selection against introgression (S8 Fig, $\rho = -1.00$, $CI_{95}[-1.00, -0.90]$).

In contrast, sympatric *mexicana* shows an unexpected negative relationship between recombination rate and introgression, with reduced maize ancestry in the highest recombination rate regions of the genome ($\rho = -1.00$, $CI_{95}[-1.00, -0.90]$). Correlations with coding bp per cM and based on $f_4$ statistics corroborate this pattern (see S6 Fig). One explanation is that some portion of maize alleles are beneficial in a *mexicana* background. While maize ancestry

in general is not predicted to provide adaptive benefits in teosinte, invasive *mexicana* in Europe shows selective sweeps for maize ancestry at multiple loci that have contributed to its establishment as a noxious weed [64] and we speculate that maize could be a source of alleles adapted to human-modified landscapes.

We repeated these analyses using local ancestry calls as our introgression estimates and found a non-significant Spearman's rank correlation between *mexicana* introgression and recombination rates for 1 cM windows in sympatric maize (S9 Fig, $\rho = 0.011$, $CI_{95}[-0.038, 0.061]$) and a positive rank correlation between maize introgression and recombination rate in sympatric *mexicana* ($\rho = 0.385$, $CI_{95}[0.341, 0.428]$). Contrasting results between global and local ancestry methods could be a reflection of true evolutionary differences across different time periods; local ancestry methods capture patterns from more recent gene flow that comes in longer ancestry blocks while STRUCTURE-like algorithms (NGSAdmix) and $f_4$ statistics are based on allele frequencies that collapse information across ancestry blocks of any size, capturing a longer evolutionary time scale. This interpretation would suggest that *mexicana* has experienced stronger selection against more recent maize gene flow than historical gene flow. However, we caution that local ancestry methods may also have subtle biases in power that are sensitive to local recombination rates and make them less reliable for comparing ancestry patterns across recombination rate quintiles.

Overall, we find support for widespread selection against introgression into maize and mixed results from similar tests of this hypothesis in *mexicana*.

## High introgression peaks shared across populations

To assess adaptive introgression in our sympatric populations, we identified introgression 'peaks' where minor ancestry exceeds the genomewide mean by more than 2 standard deviations. We find no strong reduction in average diversity ($\pi$) for *mexicana* ancestry at high introgression peaks (S4 Fig). This maintenance of diversity implies that selection at most peaks has favored multiple *mexicana* haplotypes, and hard sweeps for recent beneficial mutations on a specific haplotype are rare.

We observe that many high *mexicana* ancestry peaks are shared across subsets of our 14 maize populations (see e.g. chr4, Fig 5). While most outlier peaks are unique to a single population, many peaks are shared across 7 or more of the populations (S10A Fig). To a lesser extent, we also observe sharing of high-introgression peaks for maize ancestry in sympatric *mexicana* populations (S10B Fig).

High introgression peaks in many independent populations would be very unexpected by chance. However, our sampled populations do not provide independent evidence for adaptive introgression, due to shared gene flow and drift post-admixture (e.g. long-distance human-assisted dispersal of maize seed). To estimate the rate of peak sharing we should expect from demographic processes alone, we simulated 100,000 unlinked loci under a multivariate normal distribution (MVN) parameterized with the empirical ancestry variance-covariance matrix K (see Materials and methods). These simulations preserve the ancestry variance across loci within populations and non-independence in ancestry between populations.

For both sympatric maize and *mexicana*, every population shares an excess of high introgression peaks with all other populations compared to expectations set by our MVN null model. However, peak sharing is most elevated among high elevation maize populations (with the exception of Cocotilan, see Fig 6). To investigate the origins of population-specific peaks of introgression, we calculated $F_{ST}$ between homozygous *mexicana* ancestry in local maize and their paired local *mexicana* population for these genomic regions. Patterns of $F_{ST}$ between local sympatric pairs at local introgression peaks differed little from background $F_{ST}$ (S30 Fig),

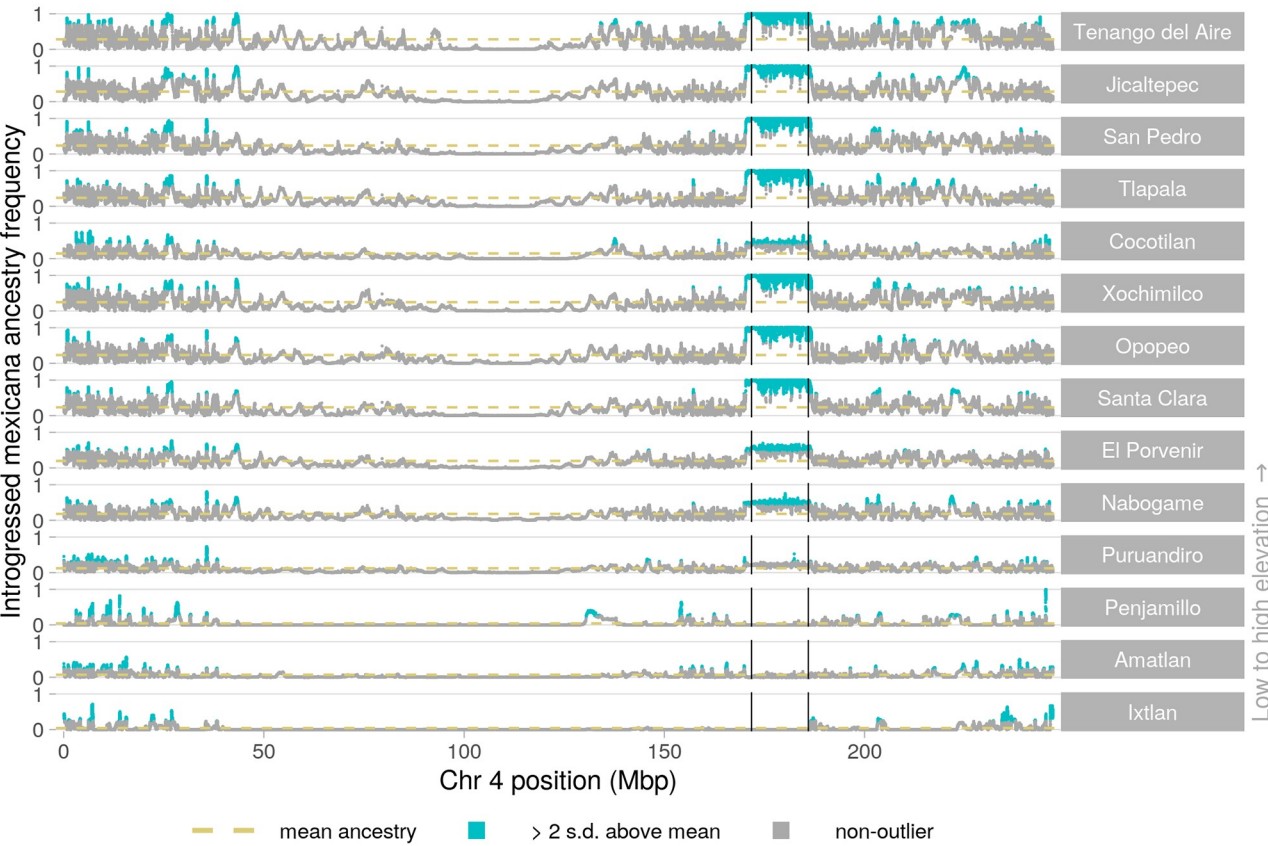

**Fig 5. Introgression in maize populations across chromosome 4.** Local introgressed *mexicana* ancestry frequency for each maize population compared to their genomewide mean. Populations are ordered from high to low elevation (top to bottom). High introgression peaks with more than 2 standard deviations above the population mean introgressed *mexicana* ancestry are highlighted in blue. Vertical black lines show the previously identified endpoints for a large inversion (*Inv4m*; coordinates from Fig 3 of [50]). For local ancestry on other chromosomes and for maize introgression into sympatric *mexicana*, see S11–S29 Figs.

offering little support for the idea that population-specific peaks arose from recent, locally sourced, adaptive introgression. Instead, patterns in maize are consistent with introgressed *mexicana* ancestry tracts from old shared admixture being favored by natural selection, and thus rising to high frequency, in a subset of populations.

This lack of local adaptive introgression is perhaps surprising given the genetic structure in *mexicana* associated with different ecotypes [49] and evidence for local adaptation within teosinte across elevation [31, 50, 51]. However, *mexicana* also has substantial standing variation and we find little evidence for hard sweeps, so one possibility is that local maize and local *mexicana* are adapting to the same environment by different available genetic paths, or even the same causal SNP on a different set of haplotype backgrounds. Older introgressed tracts may also offer more accessible paths for maize adaptation, having already purged some of their linked deleterious variation. Additionally, local exitinction and re-colonization by *mexicana* is common [37] and may contribute to a lack of local sourcing of adaptive haplotypes from contemporary *mexicana* populations.

### Genomewide scan for selection on introgressed ancestry

We scanned the genome for two types of widespread selection on introgressed ancestry: consistent selection across populations creating an overall excess or deficit of introgression, and

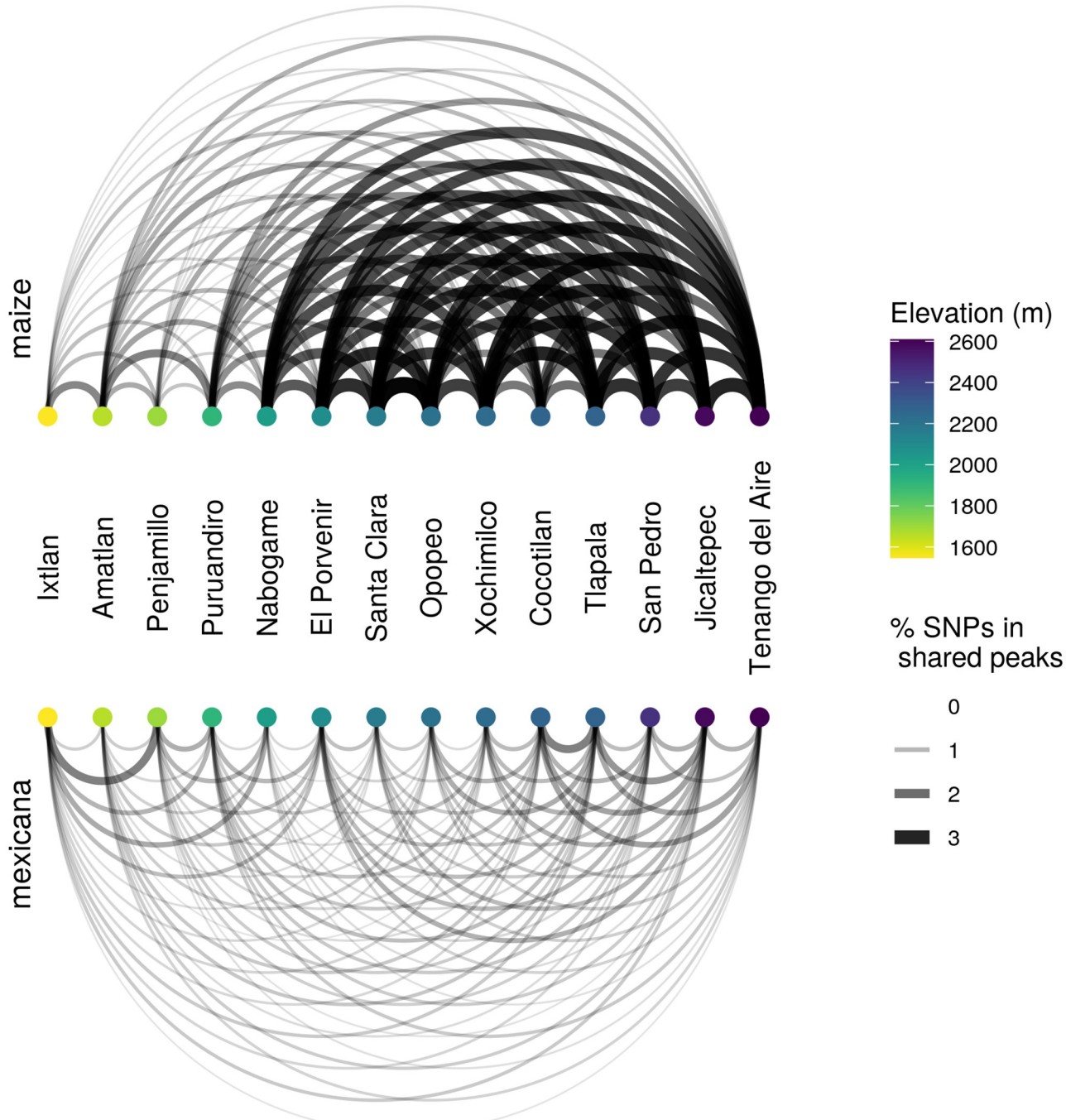

**Fig 6. Introgression peaks shared across populations.** Networks for sympatric maize (top) and *mexicana* (bottom), where each node is a sampled population labelled by location and ordered by elevation. Edges connecting a pair of populations represent the percent of SNPs within shared ancestry peaks (introgressed ancestry > 2 s.d. above each population's mean ancestry). Sharing between all pairs of populations exceeds expectations based on multivariate-normal simulations that model genomewide covariance in ancestry. The relatively darker thicker lines connecting the high elevation maize populations (except for Cocotilan), indicate that these populations share high introgression peaks at especially high frequencies.

fitness trade-offs creating steep clines in *mexicana* ancestry across elevation. We used our MVN simulated ancestry frequencies to set false-discovery-rates for excess and deficits of *mexicana* and maize introgression as well as steeper than expected slopes between *mexicana* ancestry and elevation (see S31 Fig for model fit).

We find several regions with high introgression in both directions that are unlikely to be explained by shared demographic history alone (Fig 7A). These regions of adaptive

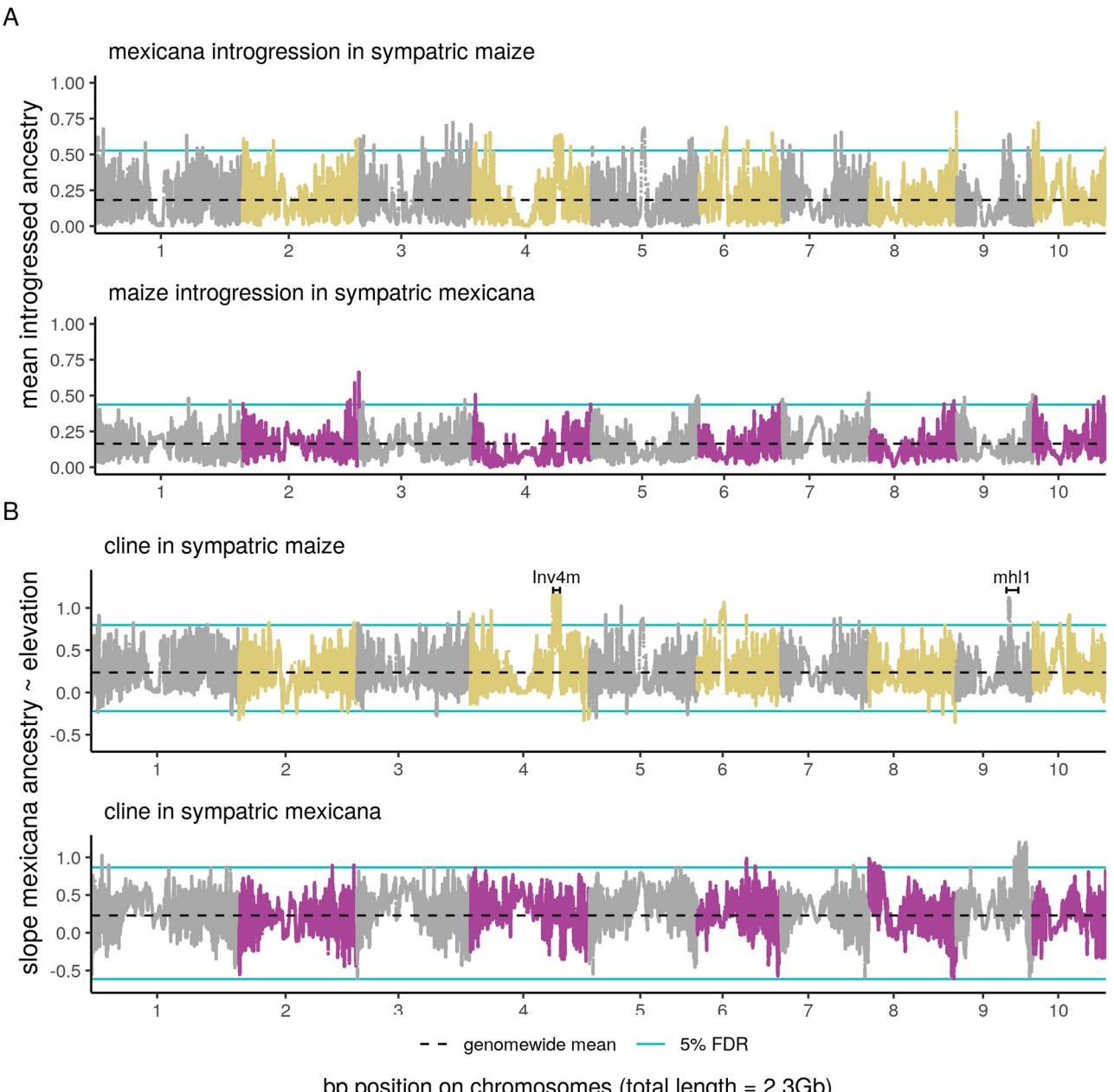

**Fig 7. Genomewide scan for selection on introgressed ancestry.** (A) Mean *mexicana* ancestry introgressed into sympatric maize populations and mean maize ancestry introgressed into sympatric *mexicana* populations. (B) Slope of *mexicana* ancestry proportion over a 1 km elevation gain in sympatric maize and *mexicana* populations. In both (A) and (B) the blue lines show the 5% false discovery rates, set using multi-variate normal simulations. Positions for *Inv4m* [50] and the *mhl1* locus [65] were converted to the maize reference genome v4 coordinates using Assembly Converter (ensembl.gramene.org). Chromosome numbers are placed at the centromere midpoint (approximate centromere positions are from [66]).

introgression ($< 5\%$ FDR) are spread across the genome and cover a small fraction ($<0.5\%$) of the genome in both subspecies. We additionally find evidence of adaptive *parviglumis* introgression into each subspecies (S32 Fig). We do not have power to determine if individual genes or regions are barriers to introgression because zero introgressed ancestry is not unusual under our simulated neutral model, given both low genomewide introgression and positive ancestry covariance between admixed populations (Fig 7).

Additionally, we identify outlier loci across the genome where *mexicana* ancestry forms steep clines across elevation (Fig 7). Our top candidate for strong associations between introgression and elevation in maize is *Inv4m*, a large 14 Mb inversion on chromosome 4 previously identified to have introgressed into high elevation maize populations [43–45, 67]. This inversion maintains steep elevational clines within teosintes [50], overlaps QTLs for leaf pigmentation and macrohairs [42], and is associated with increased yield in maize at high elevations and decreased yield at low elevations [67], but has thus far eluded functional characterization of genes within the inversion [67].

Our second strongest association co-localizes with *macrohairless1* (*mhl1*), a locus on chromosome 9 that controls macrohair initiation on the leaf blade [65] and is associated with a major QTL explaining 52% of macrohair variation between high and low elevation teosinte mapping parents [42]. Within teosintes, populations of the lowland ancestor of maize, *parviglumis*, show convergent soft sweeps at the *mhl1* locus not shared by *mexicana* [32]. Macrohairs are characteristic highland phenotypes in teosinte and maize and are thought to confer adaptive benefits through insect defence and/or thermal insulation [41, 42]. We identified a 3 Mb outlier region within the larger *mhl1* QTL which we analyzed further using PCA. We found three genetic clusters along the first principal component, evidence that an inversion polymorphism (hereafter *Inv9f*) maintains differentiation between maize/*parviglumis* and *mexicana* haplotypes across this region (S33 and S34 Figs). Additionally, we found evidence that the *mexicana*-type allele at the inversion segregates at low frequency within our lowland *parviglumis* reference population. Based on reduced diversity, the lowland maize/*parviglumis*-type allele at the inversion is likely derived (S6 Table). Thus *mexicana*-alleles at *Inv9f* could have been inherited by maize either through introgression or incomplete lineage sorting before selection pushed them to high frequency in highland populations.

The clinal patterns of admixture that we observe at inversions *Inv4m* and *Inv9f* suggest they contribute to elevation-based adaptation in maize, with variation in their fitness impacts even within the historic elevational range of *mexicana*. While our highest peaks localize with regions previously associated with characteristic highland phenotypes, many additional outlier regions with steep increases in *mexicana* ancestry across elevation have undiscovered associations with local adaptation to elevation. Additionally, outliers for steep ancestry slopes across elevation in sympatric *mexicana* suggest that introgression from maize into *mexicana* may facilitate adaptation in *mexicana* at the lower end of its elevational range.

## Selection at candidate domestication genes

We hypothesized that domestication genes will be barriers to introgression bilaterally between maize and *mexicana* [43]. While we do not have power to identify individual outlier genes that have low introgression, we can test for enriched overlap between 'introgression deserts' and a set of putative domestication genes spread across the genome.

We examined introgression for a sample of 15 well-characterized domestication genes from the literature (see S7 Table), and compared them to the regions of the genome with the lowest 5% introgression of teosinte ancestry into sympatric maize and maize ancestry into sympatric *mexicana* ('introgression deserts'). A small but enriched subset of these domestication genes

overlap with introgression deserts in sympatric maize (5, P < 0.001) and likewise in sympatric *mexicana* (5, P = 0.001).

Among these 15 domestication genes, we find that *teosinte branched1* (*tb1*), a key transcription factor that regulates branching vs. apical dominance [68, 69], overlaps introgression deserts in both maize and *mexicana*, consistent with *tb1*'s role at the top of the domestication regulatory hierarchy [70]. We also find evidence for reduced introgression into both maize and *mexicana* at *teosinte glume architecture1* (*tga1*) [71, 72], which is associated with 'naked' edible grains.

Another six domestication genes have low introgression in one direction only [73–77] (see S7 Table). Among these, *sugary1* (*su1*) in the starch pathway has low maize ancestry in *mexicana* but shows a steep increase in introgressed *mexicana* ancestry proportion with elevation in maize (+0.95 per km, < 5% FDR), which suggests this gene has pleiotropic effects on non-domestication traits in maize, with fitness trade-offs across elevation. *Sugary1* mutations modify the sweetness, nutrient content and texture of maize kernels (e.g. sweet corn), but also affect seed germination and emergence at cold temperatures [78], candidate pleiotropic effects that could be more deleterious at higher elevations.

The remaining seven domestication genes do not overlap introgression deserts in either subspecies despite evidence for their roles in domestication: *zfl2* (cob rank) [79–81], *pbf1* (storage protein synthesis) [82], *ba1* (plant architecture) [83], *ae1* (starch biosynthesis) [76], *ra1* and *ra2* (inflorescence architecture) [84, 85] and *ZmSh1–5.1+ZmSh1–5.2* (seed shattering) [75]. Despite evidence of introgression at many domestication loci, maize populations retain all of the classic domestication traits, and *mexicana* populations maintain 'wild' forms. Epistasis for domestication traits [47] could help explain this discrepancy if compensatory effects from other loci contribute to maintaining domestication traits in admixed highland maize, or if key domestication alleles segregate at moderate frequencies within *mexicana* but do not have the same phenotypic effects in a teosinte background.

## Selection within the flowering time pathway

Flowering earlier is adaptive in high-elevation environments where days are cooler and there are fewer total growing degree days in a season. We therefore expect an excess of introgressed *mexicana* ancestry at flowering time genes that may contribute to adaptive early flowering in highland maize. For example, the *mexicana* allele at *High PhosphatidylCholine 1* (*HPC1*) has recently been shown to reduce days to flowering and confer a fitness benefit in maize at higher elevations [86], and we find that the steepest clinal SNP within *HPC1* has a +0.79 *mexicana* ancestry proportion increase per km (FDR < 10%), adding support for HPC1's role in adaptive earlier flowering at higher elevations in Mexican maize varieties (S35 Fig).

We tested more broadly for enriched selection within the flowering time pathway using a set of 849 candidate flowering time genes [87, 88] and a more stringent 5% FDR cutoff. No genes from the core flowering time pathway [87] and only 13/806 other candidate flowering time genes [88] (+/- 20kb) overlap outlier regions with steep increases in *mexicana* introgression with increasing elevation in sympatric maize, which matches expected overlap by chance (1.5%, P = 0.93). Thus steep clinal introgression patterns, indicative of strong fitness trade-offs across elevation, are the exception, not the rule, for flowering-time related genes. While maladaptive flowering times have strong fitness consequences, flowering time is also a highly polygenic trait [89], which may reduce the strength of selection at most individual genes to below what we can detect using steep ancestry clines. It is alternatively possible that *mexicana* alleles show adaptive benefits across the entire range sampled (moderate to high elevation), but we

find that only 11/849 candidate flowering time genes overlap high mean *mexicana* introgression outliers at a 5% FDR (P = 0.95).

## Conclusion

Intrinsic genetic incompatibilities and partial temporal isolation (offset flowering times) create an incomplete barrier to gene flow and F1 hybrids are commonly observed in the field [37], suggesting that hybridization is frequent and ongoing. Yet, we find little evidence of the effects of recent gene flow in either direction between sympatric maize-*mexicana* population pairs. One contributing factor may be selection on later generations: while hybrids are very challenging to identify and weed out from maize fields at early life stages, farmers can easily distinguish between maize and hybrids when choosing which cobs to plant for the next season. In the other direction, hybrids in most locations are expected to be partially temporally isolated from *mexicana* and hybrid seeds that do not disarticulate are farmer-dependent for successful dispersal and reproduction, although first-generation backcrosses to *mexicana* have been observed [37].

Consistent with domestication loci acting as barriers to introgression, in both maize and *mexicana* an enriched subset of candidate domestication genes overlap 'introgression deserts.' More generally, we find introgressed *mexicana* alleles are on average deleterious in maize, but less evidence for a genomewide effect of selection against introgression into *mexicana*, possibly because epistasis masks the impact of maize alleles in a *mexicana* background [47].

Some loci show exceptional ancestry patterns consistent with selection favoring introgression in multiple populations, especially for *mexicana* ancestry in the highest elevation maize. While these shared signatures of adaptive introgression are the most striking, the majority of ancestry peaks are exclusive to a single population. Despite this signature of geographically-restricted local adaptation from *mexicana* ancestry, there is no evidence of local population sources for locally adapted haplotypes at these peaks. Indeed, timing of admixture estimates and differentiation of *mexicana* haplotypes within maize genomewide ($F_{ST}$) and at individual introgressed outlier loci (e.g. *Inv9f* at the *mhl1* locus (PCA)) suggest an ancient origin of introgressed haplotypes. We conclude that the majority of *mexicana* ancestry introgressed into maize over 1000 generations ago and has subsequently been sorted across an elevational gradient and by selection within individual populations.

## Materials and methods

### Population sampling

We used maize and *mexicana* seed accessions sampled from locations across Mexico in 2008 [43] and currently stored at UC Davis. We included 14 maize and 14 *mexicana* accessions that are paired populations sampled in sympatry from the same locations: Ixtlan*, Amatlan, Penjamillo, Puruandiro*, Nabogame*, El Porvenir*, Santa Clara*, Opopeo*, Xochimilco*, Cocotilan, Tlapala, San Pedro*, Jicaltepec and Tenango del Aire* (see S1 Table). A previous study of crop-wild admixture genotyped different maize and *mexicana* individuals from 9 of these locations (marked with *), using the Illumina MaizeSNP50 Genotyping BeadChip [43]. In addition, we chose three population accessions to sequence as a *mexicana* reference panel: Puerta Encantada and Malinalco were chosen because they have no record of contemporary maize agriculture nearby and a third population, Amecameca, was added as a complement to these two reference populations because it grows at a higher elevation, beyond the historical range of *parviglumis*.

At each sampling location, multiple ears from maternal plants were collected for seed. Population accessions varied in the number of maternal plants with viable seeds. When available,

we planted multiple seeds within each ear but only randomly selected one individual for sequencing from the plants that successfully germinated in the greenhouse.

### DNA extraction and sequencing

We extracted DNA from leaf tissue and then prepared sequencing DNA libraries using a recently published high-throughput protocol ("Nextera Low Input, Transposase Enabled protocol" [90]) with four main steps: (1) DNA shearing and tagmentation by the Nextera TD enzyme, (2) PCR amplification (Kapa2G Robust PCR kit) and individual sample barcoding (custom 9bp P7 indexing primers) (3) library normalization and pooling, and (4) bead-based clean-up and size-selection of pooled libraries. We sequenced the resulting pooled libraries using multiple lanes on Illumina HiSeq 4000 and Novaseq 6000 machines (paired-end 150 bp reads).

To address low sequencing output from some libraries, we re-sequenced 26 libraries (and merged output) and replaced 53 lower-coverage libraries with a higher-coverage library prepared from another seed grown from the same half-sibling family. We excluded 7 samples from analysis because their final libraries did not yield sufficient sequencing output (<0.05x coverage after filtering reads for mapping quality). We additionally removed one lane of sequencing (58 samples) from the study after determining a labelling error had occurred for that plate.

In total, we obtained whole genome sequences for 348 individuals (1.0x average coverage, range: 0.1–2.4x). Of these samples, 43 are *mexicana* from the three reference populations, with a total of 34.1x combined coverage. The remaining samples are maize and *mexicana* from paired sympatric populations, 262 of which have sufficient coverage for local ancestry inference ($\geq$ 0.5x, 6–12 per sympatric population, see S36 Fig). Raw sequencing reads for these low-coverage maize and *mexicana* genomes are available at NCBI (PRJNA657016).

### Reference genome and recombination map

We used version 4 of the B73 maize reference genome [66] (Zea_mays.B73_RefGen_v4.dna.toplevel.fa.gz, downloaded 12.18.2018 from Gramene).

To find local recombination rates, we converted marker coordinates from a published 0.2 cM genetic map [62] to the v4 maize genome using Assembly Converter (ensembl.gramene.org). We removed any markers that mapped to a different chromosome or out of order on the new assembly, and extended the recombination rate estimates for the most distal mapped windows to the ends of each chromosome (S37 Fig). From this map, we used *approx()* in R (v3.6.2 [91]) to estimate the cM position for any bp position, based on linear interpolation.

### Read mapping and filtering

First, we checked read quality using fastQ Screen (v0.14.0 [92]) and trimmed out adapter content from raw sequencing reads using the trimmomatic wrapper for snakemake (0.59.1/bio/trimmomatric/pe) [93]. We mapped trimmed reads to the maize reference genome using bwa mem (v0.7.17 [94]). We then sorted reads using SAMtools (v1.9 [95]), removed duplicates using picardtools (v2.7.1) MarkDuplicates and merged libraries of the same individual sequenced on multiple lanes using SAMtools merge. In all subsequent analyses in the methods below we filtered out reads with low mapping scores (< 30) and bases with low base quality scores (< 20).

## High-coverage *Tripsacum* genome sequencing

In addition to low-coverage genomes for maize and *mexicana*, we selected a *Tripsacum dactyloides* individual as an outgroup and sequenced it to high coverage. This individual is an outbred 'Pete' cultivar (rootstock acquired from the Tallgrass Prairie Center, Iowa, USA). We extracted genomic DNA from leaf tissue using the E.Z.N.A.® Plant DNA Kit (Omega Biotek), following manufacturer's instructions, and then quantified DNA using Qubit (Life Technologies). We prepared a PCR-free Truseq DNA library and sequenced it with an Illumina HiSeq2500 rapid run (paired-end 250 bp reads). We generated a total of 136.53 Gb of sequencing for this individual, available at NCBI (SRR7758238). For the following analyses that use *Tripsacum* as an outgroup, we randomly subsampled 50% of reads using seqtk, for approximately 30x coverage. We mapped reads to the maize reference using the pipeline described above, and additionally capped base quality scores with the 'extended BAQ' model in SAMtools [96], which reduces the influence of bases with lower alignment quality.

## Additional genomes from published sources

For a maize reference population, we used 55 previously published high-coverage genomes from a 'Tuxpeño' maize population grown at 983 m near Palmar Chico (NCBI: PRJNA616247 [53, 54]).

For a *parviglumis* reference population, we used 50 previously published high-coverage lowland individuals sampled from the 'Mound' population at 1,008 m near Palmar Chico [53–55] (NCBI: PRJNA616247, see S2 Table). These maize and parviglumis reference populations were sampled about 1 km apart from each other but maintain high $F_{ST}$ and are isolated by differences in flowering time [54]. Both maize and *parviglumis* reference populations are allopatric to *mexicana*, growing well below its elevational range. We mapped and filtered reads for these reference maize and *parviglumis* individuals using the pipeline described above and capped base quality scores using BAQ.

## SNP calling

We called SNPs using a combined panel of the 348 low-coverage maize and *mexicana* genomes sequenced in this study and 105 high-coverage published genomes for reference maize and *parviglumis* described above. We used ANGSD (v0.932 [97]) to identify variant sites with minor allele frequencies ≥ 5% in the total sample based on read counts ('angsd -doMajorMinor 2 -minMaf 0.05 -doCounts 1 -doMaf 8'). In addition to mapping and base quality filters ('-minMapQ 30 -minQ 20'), we capped base qualities using the extended per-Base Alignment Quality algorithm ('-baq 2' [96]) and removed sites that did not have at least 150 individuals with data or had sequencing depth exceeding 2.5x the total sample mean depth. To apply this total depth filter, we estimated mean depth ('angsd -doCounts 1 -doDepth 1 -maxDepth 10000') for 1000 regions of length 100bp randomly sampled using bedtools (v2.29.0 [98]). In total, we identified 61,612,212 SNPs on the assembled chromosomes. In conjunction with SNP calling, we produced genotype likelihoods for each individual at these variant sites using the SAMtools GL method [95] implemented in ANGSD ('-GL 1 -doGlf 2').

## Global ancestry inference

To estimate genetic relationships between populations and genomewide ancestry proportions, we used methods specific to low-coverage data that rely on genotype likelihoods, rather than called genotypes. Because these methods are sensitive to SNPs in high linkage disequilibrium (LD), we thinned genotype likelihoods to every 100th SNP (~4kb spacing) [99]. To first

confirm that maize and *mexicana* ancestry form the major axis of genetic variation in our sample, we estimated the genetic covariance matrix between all individuals using PCAngsd (v0.98.2 [100]) and visualized principal components computed using *eigen()* in R. We then estimated global ancestry proportions using the same thinned genotype likelihood files as input to NGSAdmix [56], using K = 3 clusters. Clusters clearly mapped onto the three reference groups, which we used to label the three ancestry components as 'maize', '*mexicana*' and '*parviglumis*'.

## Local ancestry and timing of admixture

We inferred local ancestry across the genome using a hidden Markov model that is appropriate for low-coverage data because it models genotype uncertainty down to the level of read counts for all admixed individuals (ancestry_hmm [58]). This method relies on allele counts from separate reference populations to estimate allele frequencies for each ancestry. Because some of our reference individuals have too low of coverage to accurately call genotypes, we randomly sampled one read per individual to get unbiased frequency estimates for major and minor alleles at each site ('angsd -doCounts 1 -dumpCounts 3'). To maximize ancestry-informativeness of sites in this analysis, we identified SNPs in the top 10% tail for any of the three possible configurations of the population branch statistic between maize, *mexicana* and *parviglumis* reference populations, thereby enriching for SNPs that distinguish one reference population from the other two. We only considered SNPs with at least 44 reference maize, 12 reference *mexicana*, and 40 reference *parviglumis* individuals with sequencing coverage at a site. For these SNPs, we estimated reference population allele frequencies from angsd ('angsd -doMajorMinor 3 -GL 1 -baq 2 -doMaf 1'), then estimated pairwise $F_{ST}$'s using the Hudson 1992 estimator [101] as implemented by Bhatia et al. 2013 [102] to calculate the population branch statistic [103] (PBS equation from pg 8 of the supplement). We then calculated genetic distances between SNPs using the maize recombination map and filtered our enriched variants to have minimum 0.001 cM spacing between adjacent SNPs.

Running ancestry_hmm jointly infers local ancestry for each individual and the timing of admixture. This HMM method assumes a neutral demographic history in which a constant-size population receives a pulse of admixture $t$ generations in the past, and finds the $t$ that maximize the likelihood of the observed read counts and hidden local ancestry state across each admixed individual's genome. The timing of admixture defines the generations for possible meiotic recombination between ancestry tracts, and therefore scales the transition probabilities between hidden ancestry states. We ran ancestry_hmm under a three-way admixture model: a population is founded 10,000 generations ago by *mexicana*, then receives a pulse of *parviglumis* ancestry at $t_{parv}$ (prior: 1000 generations ago, range: 0–10,000) and a pulse of maize ancestry at $t_{maize}$ (prior: 100 generations ago; range: 0–10,000). Because these *Zea mays* subspecies are all annual grasses, generations can equivalently be interpreted as years since admixture. In addition to $t_{parv}$ and $t_{maize}$, the HMM outputs the posterior probabilities for homozygous maize, homozygous *mexicana*, homozygous *parviglumis*, and heterozygous ancestry for each individual at every site. We analysed each sympatric maize and *mexicana* population separately, using the population's mean NGSAdmix global ancestry estimate as a prior for mixing proportions, an approximate effective population size (Ne) of 10,000 individuals, genetic positions for each SNP based on the maize linkage map, an estimated sequencing base error rate of $3 \times 10^{-3}$, and the three-way admixture model described above. We ran ancestry_hmm with an optional setting to bootstrap 100 random samples of 1,000-SNP genomic blocks to estimate uncertainty around the estimated generations since admixture ($t_{parv}$ and $t_{maize}$). To test the sensitivity of the HMM to our choice of Ne, we re-ran ancestry_hmm with two other Ne's that differ by an

order of magnitude (Ne = 1k, 100k), but did not analyze these results further after finding high correspondence for both local ancestry and timing estimates (S38 Fig).

To get a single point estimate for local ancestry at a site for an individual, we computed a sum of ancestry from the different possible ancestry states, weighted by their posterior probabilities, e.g. *mexicana* ancestry proportion = P(homozygous *mexicana*) + ½P(heterozygous *mexicana*-maize) + ½P(heterozygous *mexicana-parviglumis*). In addition, for analyses that require ancestry tract positions, we assumed that the estimated ancestry at a focal site extends halfway to the next site with a local ancestry estimate.

## Diversity within ancestry

Using local ancestry estimates from the HMM, we identified high-confidence homozygous ancestry tracts (posterior > 0.8). We filtered individual bams for reads that overlap these tracks and used the resulting filtered bams to calculate diversity within maize, *parviglumis*, and *mexicana* ancestry, separately. We estimate diversity using the ANGSD/realSFS framework which is appropriate for low-coverage sequence data it takes into account uncertainty in both genotypes and variant sites. We created a concensus fasta sequence for *Tripsacum* ('angsd -doFasta 2') to use as the ancestral state for polarizing the unfolded site frequency spectrum in these analyses.

For each population and ancestry, we estimated the site allele frequencies ('angsd -doSaf 1 -GL 1') and subsequently estimated the genomewide site frequency spectrum (SFS). We then used this SFS as a prior to estimate within-ancestry pairwise diversity ($\pi$) genomewide from the site allele frequencies ('realSFS saf2theta').

For each pair of populations and ancestry, we additionally used realSFS to estimate the two dimensional SFS from the individual population site allele frequencies genomewide. We then used this 2D SFS as a prior to estimate genomewide within-ancestry $F_{ST}$ between the two populations ('realSFS fst index -whichFst 1'). This call uses Hudson's $F_{ST}$ estimator [101] as parameterized in [102].

## Effect of local recombination rate on introgressed ancestry

To estimate the effects of linked selection and recombination rate on genomewide introgression patterns, we compared introgressed ancestry estimates across genomic quintiles. Based on a 0.2 cM-resolution recombination map [62] for maize, we merged adjacent recombination windows into larger 1 cM non-overlapping windows and calculated each window's mean recombination rate and overlap with coding base pairs (bedr 'coverage') [104]. We retrieved coding regions ('CDS') using gene annotations from Ensembl (ftp://ftp.ensemblgenomes.org/pub/plants/release-41/gff3/zea_mays/Zea_mays.B73_RefGen_v4.41.chr.gff3.gz, dowloaded 11.6.2018). We sorted windows into quintiles for either recombination rate or coding density (bp/cM). Each quintile covers approximately ⅕ of the genome based on physical bp.

## i. NGSAdmix estimates

To estimate ancestry proportions for each recombination rate quintile, we first reduced LD by thinning to 1% of SNPs (every 100th) and ran NGSAdmix 5 times separately (once per quintile) with K = 3 clusters. We assigned 'maize', '*mexicana*' and '*parviglumis*' labels to the ancestry clusters based on majority assignment to the respective reference panels.

To bootstrap for uncertainty, we re-sampled 1 cM windows with replacement from each quintile 100 times, and re-ran NGSAdmix on the resulting bootstrap SNP sets. Genetic clusters could be unambiguously assigned to maize, *parviglumis*, and *mexicana* ancestries in all but 2 bootstrap replicates for the lowest recombination quintile and 3 for the highest coding density

quintile, so we dropped this small number of replicates with unclear ancestry assignment from the bootstrap analysis. We calculated 95% percentile bootstrap confidence intervals for the estimated admixture proportions, and the Spearman's rank correlation between the recombination rate (or coding bp per cM) and admixture proportion ranks for each quintile.

We also tested for a difference in ancestry slopes with elevation across different recombination rate quintiles by fitting a linear model with an elevation by recombination quintile interaction term: *mexicana* ancestry $\sim$ elevation + r + elevation*r. Using lm() in R, we fit this model for sympatric maize and sympatric *mexicana* separately, treating quintiles as a numeric scale 0–4.

### ii. $f_4$ estimates

In a complementary analysis, we used a ratio of $f_4$ statistics as an alternative method to estimate ancestry proportions by quintile. The $f_4$ statistic measures shared genetic drift (allelic covariance) between populations in a phylogeny, due to either shared branch lengths or admixture events in the evolutionary history relating these populations. Excess shared drift with one population from a pair of sister populations in the tree is a signature of admixture, analogous to the ABBA-BABA test [105], and a ratio of two $f_4$ statistics can be used to quantify the admixture proportion. Assuming the basic phylogenetic tree relating our reference populations (((*parviglumis*, maize), *mexicana*), *Tripsacum*) in S5 Fig, we can estimate $\alpha$, the proportion of ancestry inherited from *mexicana* in a focal sympatric population, as follows [105, 106]:

$$\alpha = \frac{f_4(\text{Tripsacum}, \; parviglumis; \; \text{X}, \; \text{maize})}{f_4(\text{Tripsacum}, \; parviglumis; \; mexicana, \; \text{maize})}.$$

The denominator of this statistic estimates the branch length leading to *parviglumis* and maize that separates these sister subspecies from *mexicana*; the full ratio estimates the proportion of this branch that separates sympatric population X from *parviglumis* and maize, i.e. the *mexicana* ancestry in X. Because the $f_4$ statistic is sensitive to additional unmodeled admixture within the tree, we limited our reference *mexicana* group to individuals from just one of the three reference populations (Amecameca), which showed no evidence of admixture in our global ancestry analysis (see Fig 2).

For each 1 cM window across the genome, we used ANGSD to calculate ABBA-BABA statistics from observed read counts for the 4 populations in the numerator and denominator of the $\alpha$ estimator separately ('angsd -doabbababa2 1 -remove_bads 1 -minMapQ 30 -minQ 20 -doCounts 1 -doDepth 1 -maxDepth 10000 -useLast 1 -blockSize 5000000'). From the resulting output files, we summed the negative of the ABBA-BABA numerator ('Num') and divided by the total number of included sites ('nSites') across all 1 cM windows within a quintile to get the $f_4$ statistic [107].

We then calculated the Spearman's rank correlation between the recombination rate quintiles and admixture proportion ranks for these quintiles. We calculated simple bootstrap confidence intervals for our ancestry estimates and correlations by re-sampling 1 cM windows within quintiles with replacement 10,000 times and re-calculating the $f_4$ ratios and resulting rank correlation across quintiles to construct 95% percentile confidence intervals. We repeated this analysis using quintiles based on coding bp per cM in place of recombination rate (cM/Mbp).

### iii. Local ancestry estimates

We also calculated the Spearman's rank correlation between local recombination rate (or coding bp per cM) and local ancestry proportion at the level of individual 1 cM windows. For each

window, we averaged local ancestry estimates from the HMM across all individuals within sympatric maize, and separately, sympatric *mexicana*. We then calculated simple bootstrap confidence intervals for our local ancestry estimates and local recombination rate (or coding bp per cM) by re-sampling 1 cM windows across the genome with replacement 10,000 times and re-calculating the rank correlation across windows to construct 95% percentile confidence intervals.

### Local ancestry simulations

We simulated maize, *mexicana* and *parviglumis* ancestry population frequencies marginally using a multivariate-normal null model, e.g.

$$mexicana \text{ ancestry} \sim \text{MVN}(\vec{\alpha}, K)$$

where $\vec{\alpha}$ is the vector of mean *mexicana* ancestry frequencies genomewide for each sympatric population and $K$ is the marginal *mexicana* ancestry variance-covariance matrix relating these 14 populations and estimated from the empirical data. The diagonal entries of the K matrix capture the expected variation in local ancestry across the genome within populations due to drift and random sampling. The off-diagonals capture ancestry covariances between populations created by shared gene flow and drift post-admixture: at loci where one population has an excess of introgression, other admixed populations with shared demographic history will also tend to have an excess of introgression.

To construct K, we calculated the covariance in ancestry between each pair of populations $i$ and $j$ using all $L$ loci with local ancestry calls genomewide:

$$K[i,j] = \frac{1}{L} \sum_{l=1}^{L} (Anc_{i,l} - \alpha_i)(Anc_{j,l} - \alpha_j).$$

Above, $Anc_{i,l}$ and $Anc_{j,l}$ are local ancestry frequencies at a locus $l$ while $\alpha_i$ and $\alpha_j$ are the mean local ancestry frequencies across the genome for populations $i$ and $j$.

For sympatric maize and sympatric *mexicana* separately, we calculated the empirical K matrix between populations from all 14 sympatric locations, and then took 100,000 independent draws from their MVN distribution, thereby simulating each focal ancestry marginally for all populations at 100,000 unlinked loci. Because ancestry frequencies are bounded at [0, 1] but normal distributions are not, we truncated any simulated values outside of this range.

### Introgression peaks shared between populations

To characterize introgression peak sharing between individual populations, we defined 'ancestry peaks' as sites where a population has over 2 standard deviations more introgressed ancestry than the genomewide mean. We counted the number of peaks that are shared between all pairs and combinations of populations. To compare these results to our null model, we also counted the number of introgression peaks shared by populations in our simulated dataset, using the 2 s.d. cutoff set by the empirical data to define peaks.

Because *mexicana* ancestry shows significant diversity, we additionally characterized diversity for *mexicana* ancestry peaks introgressed into maize. For all introgressed ancestry outlier regions in a focal maize population, we used ANGSD to estimate pairwise diversity within the population ($\pi$) and differentiation ($F_{ST}$) between the focal sympatric maize populations and their local sympatric *mexicana* population. We focused on the *mexicana* ancestry within peaks by limiting our diversity estimates to only include high-confidence homozygous *mexicana* ancestry tracts (posterior > 0.8). For these analyses, we pooled information across outlier

peaks, but distinguish between introgression peaks exclusive to the focal population and introgression peaks shared between the focal population and at least 3 other sympatric maize populations. We used global estimates of the SFS and 2D SFS as priors to estimate $\pi$ and $F_{ST}$ for the subsets of the genome within introgression peaks, and otherwise followed the same methods listed above in 'Diversity within ancestry'.

### Genomewide scan for ancestry outliers

For sympatric maize and *mexicana* separately, we calculated the mean introgressed ancestry across all individuals at a locus, and fit a linear model using lm() in R to estimate the slope of *mexicana* ancestry frequencies for sympatric populations across elevation (km): *mexicana* ancestry $\sim$ elevation. We then repeated these summary statistics for every locus with an ancestry call in the empirical data and each simulated locus in the MVN simulated data.

We calculated 5% false-discovery-rate (FDR) cutoffs for high and low introgressed ancestry using the Benjamini-Hochberg method [108] and simulation results to estimate the expected frequency of false-positives under our null MVN model (one-tailed tests). We repeated this approach to identify outlier loci with steep positive (or negative) slopes for *mexicana* ancestry across elevation at a 5% FDR.

### Test for reduced introgression at domestication genes

To test whether domestication genes are unusually resistant to introgression, we first defined 'introgression deserts' as regions with the lowest 5% genomewide of teosinte introgression into sympatric maize (or, separately, maize introgression into sympatric *mexicana*). We then looked up v4 coordinates on Ensembl.org for genes associated with maize domestication in the literature (S7 Table), and used bedtools 'intersect' to identify which of these genes ±20 kb overlap introgression deserts. To test for significance, we randomly shuffled the gene positions across the genome (bedtools 'shuffle') 1000 times and re-calculated overlap with introgression deserts for each permuted data set.

### Test for selection within the flowering time pathway

We identified a list of 48 core flowering time pathway genes from the literature [87], and a broader list of 905 flowering time candidate genes [87, 88]. From the combined set, we included 849 total genes (43 core pathway) which we were able to localize on assembled autosomes of the v4 reference genome using MaizeGDB gene cross-reference files [109]. We counted the number of genes ±20 kb that intersected with outlier regions for steep increases in *mexicana* introgression with elevation (and, separately, high *mexicana* introgression) in sympatric maize populations ($< 5\%$ FDR) using bedtools 'intersect', then tested for significance by repeating this analysis with 1000 randomly shuffled gene positions.

### Analysis pipeline and data visualization

We constructed and ran bioinformatics pipelines using snakemake (v.5.17.0 [110]) within a python conda environment (v3.6). We analyzed and visualized data in R (v3.6.2 [91]) using the following major packages: tidyverse (v1.3.0 [111]), viridis (v0.5.1 [112]), bedr (v1.0.7 [104]), boot (v.1.3_25 [113, 114]), gridExtra (v2.3 [115]), ggupset (v0.3.0 [116]) and tidygraph (1.2.0 [117]). All scripts can be found on our gitHub repository, https://github.com/ecalfee/hilo, which also includes a full list of software and versions (see envs/environment.yaml).

## Supporting information

**S1 Table. Population metadata.**
(CSV)

**S2 Table. Parviglumis SRA IDs.**
(CSV)

**S3 Table. Spearman's rank correlation between genomewide admixture proportions (NGSAdmix) and recombination rate (or coding bp per cM) quintiles.**
(PDF)

**S4 Table. *Mexicana* ancestry by elevation and recombination rate quintile.** Best-fitting linear models for ancestry proportion predicted by an elevation by recombination rate interaction: *mexicana* ancestry $\sim$ elevation + r + elevation*r. Here, r is the recombination rate quintile, treated as numeric [0-4]. This model only uses ancestry estimates for sympatric individuals and is fit separately for maize and *mexicana* samples.
(PDF)

**S5 Table. Spearman's rank correlation between mean local ancestry and recombination rate (or coding bp per cM) at 1 cM genomic window resolution.** Confidence intervals are constructed using the percentile method and 10,000 bootstrap replicates created by randomly re-sampling 1 cM windows within quintiles.
(PDF)

**S6 Table. Diversity within maize and *mexicana* alleles at *inv9f*.** Pairwise diversity ($\pi$) and Watterson's theta ($\theta_W$) for samples clustering by PCA with the maize-allele at the inversion (PC1 $> 0.2$) or *mexicana*-allele at the inversion (PC1 $< 0.4$). Individuals heterozygous for the inversion (or ambiguous in PCA clustering) were excluded. Diversity estimates within the putative inversion region were calculated for each group using the same ANGSD/realSFS pipeline as genomewide diversity estimates. Only subspecies with $> 5$ samples in a cluster were analysed.
(PDF)

**S7 Table. Domestication genes and overlap with introgression deserts.**
(PDF)

**S1 Fig. PCA of maize, *mexicana* and *parviglumis*.** First and second principal components from the genomewide genetic covariance matrix relating all maize, *mexicana* and *parviglumis* individuals. PC1 aligns with a maize to *mexicana* ancestry gradient while PC2 separates out *parviglumis* ancestry from the other two *Zea* subspecies.
(TIF)

**S2 Fig. $F_{ST}$ between *parviglumis* ancestry tracts from different populations.** Pairwise $F_{ST}$ between parviglumis ancestry tracts from population 1 (x-axis) and population 2 (y-axis). Populations are sorted by subspecies, then elevation. Local sympatric maize-*mexicana* population pairs are highlighted with a white dot and do not show reduced $F_{ST}$ within *parviglumis* ancestry relative to other (non-local) maize-*mexicana* comparisons.
(TIF)

**S3 Fig. Time since admixture.** Estimated generations since admixture under a three-way admixture model: a founding *mexicana* population receives a pulse of *parviglumis* admixture and then a second pulse of maize admixture (possibly resulting in majority maize ancestry). Each sympatric maize and *mexicana* population was analyzed separately under this model,

with 95% percentile confidence intervals for each admixture pulse based on 100 bootstrap samples of genomic blocks (1,000 SNPs per block). Estimates and bootstraps were produced during ancestry_hmm model fitting for local ancestry inference. Populations are ordered left to right by increasing elevation. Populations with very small genetic contributions from an ancestry pulse ($< 10\%$) are faded because their timing estimates are less certain.
(TIF)

**S4 Fig. Diversity ($\pi$) within ancestry.** Each point summarises pairwise genetic diversity ($\pi$) for genomic regions with high-confidence homozygous maize or *mexicana* ancestry, calculated separately for the sympatric maize and *mexicana* populations at each sampled location. For maize ancestry (top), only a genomewide $\pi$ is estimated, using all regions with high-confidence homozygous maize ancestry. For *mexicana* ancestry (bottom), $\pi$ is calculated and plotted separately for three subsets of the genome: introgression peaks ($> 2$ s.d. above the mean) found in the focal maize population only, introgression peaks shared between the focal maize and at least 3 other maize populations, and a genomewide estimate.
(TIF)

**S5 Fig. Population tree.** Phylogenetic tree assumed when estimating the ratio of $f_4$ statistics. The pink branch represents the shared drift between maize and *parviglumis* that is introduced to the focal sympatric population via admixture of proportion $1 - \alpha$. We used only plants from the Amecameca site in our *mexicana* reference group for this analysis because that site showed no evidence of previous admixture.
(TIF)

**S6 Fig. $f_4$ ancestry by recombination rate and gene density.** *Mexicana* ancestry proportions by genomic quintiles in sympatric maize (top) and *mexicana* (bottom), estimated using $f_4$ ratios. Spearman's rank correlations for each plot: (A) *Mexicana* ancestry by recombination rate quintile in sympatric maize ($\rho = 1.00$, $CI_{95}[0.30, 1.00]$), (B) *mexicana* ancestry by gene density (coding bp/cM) quintile in symaptric maize ($\rho = -1.00$, $CI_{95}[-1.00, -0.40]$), (C) *Mexicana* ancestry by recombination rate quintile in sympatric *mexicana* ($\rho = 1.00$, $CI_{95}[0.70, 1.00]$), and (D) *mexicana* ancestry by gene density (coding bp/cM) quintile in symaptric *mexicana* ($\rho = -1.00$, $CI_{95}[-1.00, -0.90]$). Mean ancestry per quintile and 95% percentile bootstrap confidence interval (n = 10,000) are depicted in black. Violin plots show the density of ancestry estimates for individual bootstraps re-sampled within quintiles. Note: Ancestry estimates from $f_4$'s are less reliable for sympatric *mexicana* than sympatric maize because of significant *parviglumis* ancestry in some sympatric *mexicana* populations. The $f_4$ ratio estimation method assumes no additional unmodeled admixture on the population tree (see S5 Fig).
(TIF)

**S7 Fig. Introgressed ancestry by recombination rate.** Inferred introgressed ancestry in reference populations (top) and sympatric maize and *mexicana* populations (bottom) using NGSAdmix (K = 3) by recombination rate quintiles. Group mean and 95% percentile bootstrap confidence interval (n = 100) are shown. Different colors distinguish the different introgressing ancestries, and different shapes distinguish the *Zea* subspecies for the sampled populations.
(TIF)

**S8 Fig. Introgressed ancestry by coding bp per cM.** Inferred introgressed ancestry in reference populations (top) and sympatric maize and *mexicana* populations (bottom) using NGSAdmix (K = 3) by coding density quintiles. Group mean and 95% percentile bootstrap confidence interval (n = 100) are shown. Different colors distinguish the different

introgressing ancestries, and different shapes distinguish the *Zea* subspecies for the sampled populations.
(TIF)

**S9 Fig. Local introgressed ancestry in 1 cM windows by recombination rate.** Estimated local ancestry in sympatric maize and *mexicana* samples using ancestry_hmm. Each point is a 1 cM genomic window and the line shows the best linear model fit for mean introgressed ancestry by recombination rate on a log scale. Different colors distinguish the different introgressing ancestries, and different shapes distinguish the *Zea* subspecies for the sampled sympatric populations.
(TIF)

**S10 Fig. High introgression peaks shared across sympatric populations.** Here we show the 75 most common combinations of populations that share ancestry peaks (introgressed ancestry > 2 s.d. above each population's mean ancestry) for (A) sympatric maize and (B) sympatric *mexicana*. Bar height represents the percent of SNPs genomewide within peaks shared by the populations highlighted in blue below. Only for sympatric maize do we find that larger sets of populations (brighter colored bars) commonly share peaks across the genome. Populations are ordered from high to low elevation (top to bottom), showing that introgression peak sharing is more common among combinations of the highest elevation maize populations.
(TIF)

**S11 Fig. Introgression in maize populations across chromosome 1.**
(TIF)

**S12 Fig. Introgression in maize populations across chromosome 2.**
(TIF)

**S13 Fig. Introgression in maize populations across chromosome 3.**
(TIF)

**S14 Fig. Introgression in maize populations across chromosome 5.**
(TIF)

**S15 Fig. Introgression in maize populations across chromosome 6.**
(TIF)

**S16 Fig. Introgression in maize populations across chromosome 7.**
(TIF)

**S17 Fig. Introgression in maize populations across chromosome 8.**
(TIF)

**S18 Fig. Introgression in maize populations across chromosome 9.**
(TIF)

**S19 Fig. Introgression in maize populations across chromosome 10.**
(TIF)

**S20 Fig. Introgression in *mexicana* populations across chromosome 1.**
(TIF)

**S21 Fig. Introgression in *mexicana* populations across chromosome 2.**
(TIF)

**S22 Fig. Introgression in *mexicana* populations across chromosome 3.**
(TIF)

**S23 Fig. Introgression in *mexicana* populations across chromosome 4.** Vertical lines indicate the coordinates for *Inv4m*.
(TIF)

**S24 Fig. Introgression in *mexicana* populations across chromosome 5.**
(TIF)

**S25 Fig. Introgression in *mexicana* populations across chromosome 6.**
(TIF)

**S26 Fig. Introgression in *mexicana* populations across chromosome 7.**
(TIF)

**S27 Fig. Introgression in *mexicana* populations across chromosome 8.**
(TIF)

**S28 Fig. Introgression in *mexicana* populations across chromosome 9.**
(TIF)

**S29 Fig. Introgression in *mexicana* populations across chromosome 10.**
(TIF)

**S30 Fig. Differentiation ($F_{ST}$) between introgressed ancestry tracts and local *mexicana*.** Each point summarises $F_{ST}$ between *mexicana* ancestry tracts within a focal maize population and *mexicana* ancestry tracts within the local *mexicana* population sampled at the same site. Within-*mexicana* ancestry $F_{ST}$ is presented separately for three subsets of the genome: introgression peaks found in the focal maize population only, peaks shared between the focal maize and at least 3 other maize populations, and a genomewide estimate. Notably, peaks where adaptive introgression is limited to the local population ('1 population peaks') do not have reduced $F_{ST}$ to local *mexicana* haplotypes.
(TIF)

**S31 Fig. Quantile comparison of observed data vs. MVN normal null model.** (A) QQ-plot of simulated vs. observed mean ancestry at individual loci across all sympatric individuals. (B) QQ-plot of simulated vs. observed slopes from the linear model *mexicana* ancestry $\sim$ elevation at individual loci.
(TIF)

**S32 Fig. Genomewide scan for selection on *parviglumis* ancestry.** Mean introgressed *parviglumis* ancestry into sympatric maize and *mexicana* populations populations. The blue line shows the 5% false discovery rate for high introgression, set using multi-variate normal simulations.
(TIF)

**S33 Fig. Ancestry slope with elevation at mhl1 locus.** Slope of introgressed *mexicana* ancestry proportion in sympatric maize over a 1 km gain in elevation, zoomed in on the mhl1 QTL region on chromosome 9. Coordinates for the contiguous 3 Mb outlier region within this QTL are 9:108615836-111785557.
(TIF)

**S34 Fig. PCA of putative mhl1 inversion.** Principal components analysis of all SNPs in the 3 Mb outlier region within the mhl1 QTL region that shows a steep increase in introgressed

*mexicana* ancestry across elevation (<5% FDR). This region on chromosome 9 is a putative inversion (9:108615836-111785557), separating out into three clusters across PC1: individuals homozygous for the common *mexicana* inversion allele (left), heterozygous individuals (middle) and individuals homozygous for the common maize inversion allele (right). There is evidence that the *mexicana* inversion allele segregates at low frequency in lowland *parviglumis*, but not lowland maize (right cluster includes all low-elevation reference maize). Additionally, PC2 primarily separates out diversity within the common *mexicana* allele for the inversion, with maize samples tending to have lower PC2 values.
(TIF)

**S35 Fig. Introgression patterns at *HPC1* flowering time gene.** (A) Slope of *mexicana* ancestry across elevation near the *HPC1* gene, with 1%, 5% and 10% FDRs for steep slopes indicated in blue. (B) Mean *mexicana* introgression proportions for each population, and fitted slope across elevation, for the top outlier SNP within *HPC1* (3:7737055). In higher elevation maize populations, introgression at *HPC1* far exceeds the genomewide mean *mexicana* ancestry (black circles).
(TIF)

**S36 Fig. Number of individuals sequenced per location.** Number of maize (top) and *mexicana* (bottom) individuals sequenced by this study with minimum 0.05x WGS coverage. Amecameca, Malinalco and Puerta Encantada have no paired maize samples and are used as a reference panel for *mexicana* ancestry. For sympatric maize and *mexicana*, only individuals meeting a more stringent 0.5x coverage threshold (shown in darker shading) are included in analyses based on local ancestry inference.
(TIF)

**S37 Fig. Linkage map.** (A) Over 85% of the original markers from the Ogut et al [62] 0.2cM linkage map on reference genome v2 were successfully transferred to reference genome v4, using Assembly Converter (ensembl.gramene.org). We dropped markers automatically that mapped to the wrong chromosome or in reverse map order (presumably due to small contigs having corrected orientations in the newer version of the reference genome). A small number of markers were dropped by hand for more complex out-of-order mapping errors. (B) With few exceptions, the number of markers dropped in a row is small. (C) Visualization of the full linkage map on v4 of the reference genome. Included markers are in blue while excluded markers are highlighted in orange and red. (D) Zoomed in view of all 6 regions in the genome where out-of-order markers did not form simple reversals. For these more complex regions we identified markers to drop by hand (in red) to reach a monotonically increasing map solution.
(TIF)

**S38 Fig. Sensitivity of timing estimates to choice of Ne.** Admixture timing estimates from ancestry_hmm across 3 choices of effective population size. Timing estimates are strongly correlated with slightly older estimates for the smallest Ne = 1000. Local ancestry calls across the genome, summarised within sympatric maize and sympatric *mexicana* for each Ne, were all tightly correlated ($r > 0.99$). Main results of the paper are presented only for Ne = 10,000.
(TIF)

## Acknowledgments

We thank Pesach Lubinsky for collecting the seeds sequenced in this study. We also want to thank Michael Turelli, the Coop and Ross-Ibarra labs, and the HILO and Zeavolution working groups for helpful feedback on this work.

## Author Contributions

**Conceptualization:** Erin Calfee, Graham Coop, Jeffrey Ross-Ibarra.

**Data curation:** Erin Calfee.

**Formal analysis:** Erin Calfee, Daniel Gates.

**Funding acquisition:** Erin Calfee, Graham Coop, Jeffrey Ross-Ibarra.

**Investigation:** Erin Calfee, Daniel Gates, Anne Lorant, M. Taylor Perkins.

**Methodology:** Erin Calfee, Daniel Gates, Anne Lorant, M. Taylor Perkins.

**Resources:** Graham Coop, Jeffrey Ross-Ibarra.

**Supervision:** Graham Coop, Jeffrey Ross-Ibarra.

**Visualization:** Erin Calfee.

**Writing – original draft:** Erin Calfee.

**Writing – review & editing:** Erin Calfee, Daniel Gates, Anne Lorant, M. Taylor Perkins, Graham Coop, Jeffrey Ross-Ibarra.

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
