## [Decision Letter · Decision Letter 0]

7 Sep 2021

Dear Dr Calfee,

We are pleased to inform you that your manuscript entitled "Selective sorting of ancestral introgression in maize and teosinte along an elevational cline" has been editorially accepted for publication in PLOS Genetics. Congratulations!

The reviewers did include a number of comments, but we have determined that these edits can be easily made in a final edit before final publication. We strongly encourage that you make those edits.

Yours sincerely,

Rodney Mauricio, Ph.D.

Associate Editor

PLOS Genetics

Kirsten Bomblies

Section Editor: Evolution

PLOS Genetics

Comments from the reviewers (if applicable):

Reviewer's Responses to Questions

**Comments to the Authors:**

Reviewer #1: The manuscript by Calfee et al. reports a comprehensive examination of introgression between a large set of sympatric populations of a crop (maize) and its closest wild relatives (lowland and highland teosinte subspecies). The work is well performed, well written, and yields significant new insight into forces and dynamics that have shaped the complex history and patterns of shared genetic variation among these taxa. The study is particularly notable for its inclusion and statistically rigorous assessment of shared introgression patterns across such a large number of paired populations, the finding that the bulk of the shared ancestry that even sympatric maize has with highland teosinte dates to events occurring prior to 1000 years before present, and the depth with which the authors evaluate domestication, trait, and recombination-related hypotheses about barriers or promoters of introgression important in this system. I have just a couple comments that would be helpful to address by revision.

Reporting of HMM predicted admixture tracts: The authors report summary statistics of the median and range of admixture timing values for the tracts of shared ancestry (in text or visualized in Fig. S4). However, from what I can find in the text and supplement, there is limited additional information provided about these intervals as a set. The authors should report summary statistics about the lengths of these tracts and the numbers of genes within these windows. Supplemental tables should be provided with the individual information associated these tracts and for the 1cM windows that report relevant position information, introgression proportions, recombination rates, population genomic parameters, etc. so that readers can perform additional analyses or look at specific intervals of interest.

Fig 3: The authors should add the population names along the horizontal and vertical axes so it is clear which lines correspond to Ixtlan and Penjamillo.

Reviewer #2: The manuscript, “Selective sorting of ancestral introgression in maize and teosinte along an elevational cline”, is excellent and I thoroughly enjoyed reading it. It presents a comprehensive analysis of introgression along an elevational cline between the domesticated maize and its wild highland relative mexciana. The manuscript presents an interesting example of how introgression might influence range expansion and adaptation. The maintenance of diversity in introgressed regions is a valuable contrast to other studies of introgression with strong selective sweeps. This paper is also an excellent model of different methods to parse recent versus past introgression. I have only a few minor comments related to the clarity of some of the figures and results.

The volume of supplementary material is a bit overwhelming and seems unnecessarily long for a manuscript of this scope. I think it could be streamlined to make it easier to reference. For example some of the supplementary tables list the statistics for test results (like spearman’s) that could be included in the Figure legends instead. This would be much easier than looking at 2 different supplementary files to understand these results. Likewise, some of the supplementary figures could be combined so that the differences between maize and mexicana are easier to compare and put the result in context (e.g. Fig S5 and S6; S8 and S9; S13 and S14; S17 and S16). There are also figures that convey the same information and there is no advantage to displaying the results both ways, such as Fig 2 and Fig S2, and Fig 4B and S10. Finally, there are a few mistakes in the supplement, such as Fig S7 is missing and there are some typos in the figure legends that are critical for interpreting the figures.

Minor Comments:

The abstract could be shortened

Line 48: whole genome sequences or whole genome sequencing data

Line 57: The genetic structure of Mexicana genotypes could be described in a bit more detail in one of the above paragraphs, posing the question of the spatial scale of introgression. Then here the goal of testing for ongoing local introgression could be better articulated. Right now, this objective feels like an aside in this paragraph.

Figure 1A: Is the dark green where their ranges overlap? That should be clarified in the figure caption. Indicate that maize spans the entire elevational range in the figure caption?

Figure 1B: Indicate sample sizes for each collection site? Would it be possible to use a grayscale relief map so that elevations can be visible on mountain ranges and/or relative to the overall topography?

Line 93-97: Clarify in the text that these ancestry results are for reference populations. And Lines 99-102, clarify that these are ancestry results for 14 paired sympatric populations.

Line 97: Highlight in the text that the reference population of mexicana with no parvigulumis ancestry is the high elevation site?

Figure 3: The caption description for this figure is a bit confusing. What is meant by population 1 and 2 on the X and Y axes isn’t clear.

Line 167: It might be worth mentioning somewhere in the manuscript the generation time of maize and mexicana. They both have an annual life cycle (i.e. including in wild populations)?

Line 153: These populations aren’t labeled on Figure 3.

Line 195: Could the stats for the correlations between recombination rate quintile and f4 (Fig S8) or gene density and f4 (Fig S9) just be included in the figure legend instead of a separate table (Table S4)?

Line 568: It would be helpful to provide explanation or justification for the three-way admixture model

Fig S4: are the lighter and darker colors supposed to represent pulse size? If so, that isn’t easy to understand based on the black and light gray legend symbols. It might be clearer to just explain that in the figure legend.

Fig S6: Should the legend say homozygous maize ancestry, not mexicana? If this isn’t a typo, then I am missing what is different between Fig S5 and S6. It might also be better to have the x-axis be the same range of values and to plot these figures together as panels?

Fig S7 is missing from the Supporting Figures file.

Fig S13 and S14: Are these figure legends also mixed up? Should S13 be “f4 ancestry in mexicana…” and should both be “estimated maize ancestry in sympatric Mexicana samples…”.

Reviewer #3: Review of “Selective sorting of ancestral introgression in maize and teosinte along an elevational cline”

This paper explores how introgression has played a role in the evolution of maize and its wild relative teosinte. They use low coverage whole genome sequencing across an elevational gradient for both species and find that introgression is within the last 1000 generations and that introgression in maize is more common in regions of high recombination. Furthermore, they show that there is an elevational gradient in introgression quantity and they explore whether this is likely due to selection. Lastly, they identify introgression outliers, including two inversions.

This paper, like others from the lead author, is challenging to review because it is very well written and thorough, so there are no obvious problems to comment on. The domestication and introgression history of maize with parviglumis and mexicana has been explored previously but was still not completely clear. This paper does an excellent job quantifying the timing and location of introgression, as well as getting at how selection is acting. The authors also explore the possibility of adaptive introgression through shared introgression peaks but come up against the problem of non-independence between populations. They do a very clever trick of using a model accounting for the empirical ancestry variance to get a rigorous null to compare the actual results. I’m particularly impressed by how thorough the analyses are, including testing assumptions like effective population size during analyses, and using bootstraps and permutations to test significance in different tests. The figures are clean and clear, and there are ample supplementary figures. I think this paper should be accepted as is.

Minor comments:

-Line 340: At first reading of this paragraph, it sounds like you first looked at 15 loci for overlaps with introgression deserts, then looked at a subset of 5 of them (i.e. two different enrichment tests, because the first set wasn’t enriched). This isn’t the case, but the wording a bit ambiguous and should be changed.

-Figure S7 is not included in the sup files.

-Figure S11 and S12: Individuals are not included in these figures, as described in figure caption, only mean and confidence intervals.

-Page 16. There are two sections with i. and ii., which aren’t obviously bigger section headings. Perhaps this is a formatting error, or maybe “f4 estimate” section should also have a roman numeral.

**Have all data underlying the figures and results presented in the manuscript been provided?**

Reviewer #1: **No: **See comments.

Reviewer #2: Yes

Reviewer #3: Yes

PLOS authors have the option to publish the peer review history of their article (what does this mean?). If published, this will include your full peer review and any attached files.

Reviewer #1: No

Reviewer #2: No

Reviewer #3: No

**Data Deposition**

http://datadryad.org/submit?journalID=pgenetics&manu=PGENETICS-D-21-01078

**Press Queries**

---

## [Editor Report · Acceptance letter]

6 Oct 2021

PGENETICS-D-21-01078 

Selective sorting of ancestral introgression in maize and teosinte along an elevational cline 

Dear Dr Calfee, 

We are pleased to inform you that your manuscript entitled "Selective sorting of ancestral introgression in maize and teosinte along an elevational cline" has been formally accepted for publication in PLOS Genetics! Your manuscript is now with our production department and you will be notified of the publication date in due course.

With kind regards,

Amy Kiss

PLOS Genetics

On behalf of:
